# Offline Reinforcement Learning via Tsallis Regularization

**Lingwei Zhu †**                                                    *lingwei4@ualberta.ca*
*Department of Computing Science, University of Alberta*
*Alberta Machine Intelligence Institute (Amii), Canada*

**Matthew Schlegel †**                                               *mkschleg@ualberta.ca*
*Department of Computing Science, University of Alberta*
*Alberta Machine Intelligence Institute (Amii), Canada*

**Han Wang**                                                         *han8@ualberta.ca*
*Department of Computing Science, University of Alberta*
*Alberta Machine Intelligence Institute (Amii), Canada*

**Martha White**                                                     *whitem@ualberta.ca*
*Canada CIFAR AI Chair*
*Department of Computing Science, University of Alberta*
*Alberta Machine Intelligence Institute (Amii), Canada*

**Reviewed on OpenReview:** *https://openreview.net/forum?id=HNqEKZDDRc*

## Abstract

Offline reinforcement learning (RL) focuses on learning a good policy from a fixed dataset. The dataset is generated by an unknown behavior policy through interactions with the environment and contains only a subset of the state-action spaces. Standard off-policy algorithms often perform poorly in this setting, suffering from errroneously optimistic values incurred by the out-of-distribution (OOD) actions not present in the dataset. The optimisim cannot be corrected as no further interaction with the environment is possible. Imposing divergence regularization and in-sample constraints are among the most popular methods to overcoming the issue by ensuring that the learned policy stays close to the behavior policy to minimize the occurrence of OOD actions. This paper proposes Tsallis regularization for offline RL, which aligns the induced *sparsemax* policies to the in-sample constraint. Sparsemax interpolates existing methods utilizing hard-max and softmax policies, in that only a subset of actions contributes non-zero action probability as compared to softmax (all actions) and hard-max (single action). We leverage this property to model the behavior policy and show that under several assumptions the learned sparsemax policies may have sparsity-conditional KL divergence to the behavior policy, making Tsallis regularization especially suitable for the Behavior Cloning methods. We propose a novel actor-critic algorithm: Tsallis Advantage Weighted Actor-Critic (Tsallis AWAC) generalizing AWAC (Nair et al., 2021) and analyze its performance in standard Mujoco environments. Our code is available at `https://github.com/lingweizhu/tsallis_regularization`.

## 1 Introduction

Reinforcement learning (RL) has achieved impressive successes in various domains through learning from online interactions with the environment (Mnih et al., 2015; Silver et al., 2017; Andrychowicz et al., 2020). However, online RL is often less suited to real-world domains, especially when acting unconstrained in an environment can be expensive or dangerous. Offline RL instead addresses the problem of learning good

---

† These authors contributed equally to this work.

policies completely from a given dataset generated following unknown policies. The goal of offline RL is to learn policies which outperform—or at least match—the policies used to generate the dataset.

However, many of the difficulties of offline RL stem from not using online interaction. Standard off-policy RL algorithms tend to perform poorly, due to the well-known extrapolation error or out-of-distribution (OOD) action problem: improving the learned policy beyond the level of behavior policy requires estimating values of state-action pairs not present in the dataset. Optimistic estimates will bias the agent into favoring the absent actions in the policy improvement stage, leading to a vicious loop (Fujimoto et al., 2019; Kostrikov et al., 2022). Since no further interactions with the environment is allowed, the agent may be increasingly prone to choose actions with delusively high values It is worth noting the extrapolation problem does not occur in the tabular case but rather is the result of function approximators (Gulcehre et al., 2021; Dadashi et al., 2021).

A popular branch of offline RL is behavior cloning (BC) (Pomerleau, 1988), referred to as imitation-based methods in (Xu et al., 2022). BC based methods enforce the learned policy to stay close to or reproduce the behavior policy (Dadashi et al., 2021; Ghasemipour et al., 2021; Nair et al., 2021; Siegel et al., 2020; Wang et al., 2020; Wu et al., 2022; 2020). This is often achieved in two ways: (1) by using **divergence regularization** $D(\pi_t(\cdot|s)||\pi_\mathcal{D}(\cdot|s))$ when updating policy to penalize large deviation from the learned policy $\pi_t$ to the behavior policy $\pi_\mathcal{D}$ (Brandfonbrener et al., 2021; Jaques et al., 2020; Kostrikov et al., 2021; Wu et al., 2020; Osa et al., 2023), usually $D$ is chosen to be the KL divergence but other divergences such as MMD (Kumar et al., 2019), Fisher's divergence (Kostrikov et al., 2021) or Shannon entropy (Xiao et al., 2023) have also been used; (2) imposing an **in-sample constraint** to the updates (Fujimoto et al., 2019; Kostrikov et al., 2022; Xiao et al., 2023) where the target hard max operator $\max_a Q(s,a)$ in Q-learning is replaced to the *in-sample* maximum $\max_{a:\pi_\mathcal{D}(a|s)>0} Q(s,a)$. This scheme has been recently extended to the in-sample log-sum-exp $\ln \sum_{a:\pi_\mathcal{D}(a|s)>0} e^{Q(s,a)}$ (Xiao et al., 2023).

In this paper, we propose to use a general but less studied class of regularizers that interpolates softmax and hard-max−−the Tsallis regularizers, specifically, Tsallis entropy and Tsallis KL divergence−−as the choice of $D$. While Tsallis entropy and Tsallis KL divergence have recently been investigated in online RL (Lee et al., 2018; 2020; Pacchiano et al., 2021; Zhan et al., 2023; Zhu et al., 2023), they have never been utilized in the offline RL context. Tsallis entropy (resp. Tsallis KL) is a strict generalization of Shannon entropy (resp. KL). Different from the softmax policy induced by Shannon entropy and KL that has full support (Azar et al., 2012; Kozuno et al., 2019; Vieillard et al., 2020), Tsallis regularization induces the sparsemax policy that truncates actions with low values, i.e. setting their probability to zero. We link action truncation to the in-sample constraint $\pi_\mathcal{D}(a|s) > 0$, arriving at the assumption that the behavior policy is within the sparsemax policy class and collects in the offline dataset a subset of actions with high values. Intuitively, the assumption suggests to learn a policy that shrinks the support of the behavior policy, which renders Tsallis regularization especially suited to BC or imitation-based methods. We formalize this intuition in Section 4 by showing that the KL divergence between $\pi_\mathcal{D}$ and the learned policy $\pi_t$, all within the sparsemax class, may be upper bounded depending on the Tsallis entropic index controlling the sparsity.

By combining in-sample constraint and Tsallis regularization, we propose two actor-critic algorithms: Tsallis advantage weighted actor-critic (Tsallis AWAC) based on Tsallis KL divergence; and Tsallis In-sample Actor-Critic (Tsallis InAC), based on Tsallis entropy. Both methods extend their base algorithms to the $q > 1$ domain. With the introduced sparsity by $q$-exp, Tsallis AWAC is among the best performers among the compared methods, while Tsallis InAC is less stable and only competes favorably for expert datasets.

## 2 Background

We model our problem as a Markov decision process (MDP) expressed by the tuple $(\mathcal{S}, \mathcal{A}, P, r, \gamma)$. $\mathcal{S}$ is the set of states, $\mathcal{A}$ is the set of actions. $P(\cdot|s,a)$ denotes transition probability over the state space given state-action pair $(s,a)$, and $r(s,a)$ defines the reward associated with that transition. $\gamma \in [0,1)$ is the discount factor. A policy $\pi(\cdot|s)$ is a mapping from the state space to distributions over actions. The state-action value function starting from $(s,a)$ following policy $\pi$ is defined as $Q_\pi(s,a) = \mathbb{E}_\pi \left[ \sum_{t=0}^\infty \gamma^t r(s_t, a_t)|s_0 = s, a_0 = a \right]$. In this paper we consider entropy-regularized formulation $\mathbb{E}_\pi [Q_*(s,a) - \Omega(\pi(\cdot|s))]$, where $\Omega(\pi(\cdot|s)) \in \mathbb{R}^{|\mathcal{S}|}$ is the regularizer convex in $\pi$. Popular choices for $\Omega$ include the negative Shannon entropy $-\tau \mathcal{H}(\pi(\cdot|s)) :=$

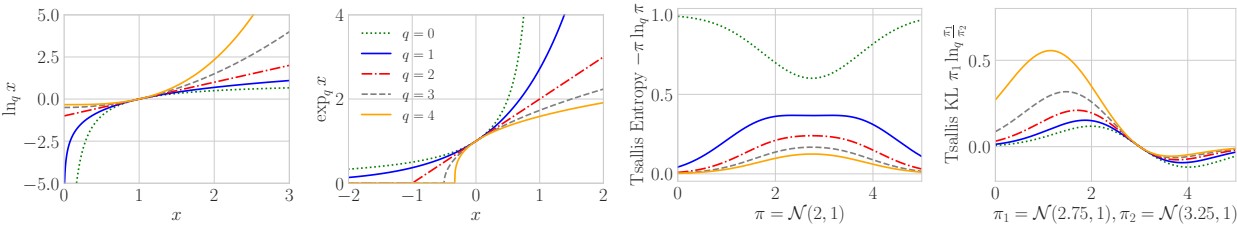

Figure 1: (From left to right) $q$-logarithm, $q$-exponential, Tsallis entropy and Tsallis KL divergence of Gaussian policies. Note that $q = 0$ is shown here only for an illustration purpose. We consider $q > 0$ in this paper for theoretically sound regularizers. When $q = 1$, the above functions recover their standard counterparts.

$\tau \sum_a \pi(a|s) \ln \pi(a|s)$, which encourages the policy to be uniform with $\tau$ weighting the effect. KL divergence $D_{KL}(\pi(\cdot|s) \| \mu(\cdot|s)) := \sum_a \pi(a|s) \ln \frac{\pi(a|s)}{\mu(a|s)}$ penalizing large deviation from the reference policy $\mu$ is another popular choice (Azar et al., 2012; Vieillard et al., 2020; Chan et al., 2022; Zhu & Matsubara, 2023).

It is worth noting that offline RL problems often have continuous action spaces, while many existing methods were derived based on a discrete footing. Approximation is usually necessary, e.g., in evaluating policies (Fujimoto et al., 2019; Xiao et al., 2023). Similar to (Xiao et al., 2023), our method is positioned in the entropy-regularized literature and established based on the discrete action setting (Vieillard et al., 2020). When applied to continuous action problems, our method also necessitates approximation in policy evaluation, as detailed in Section 5. While such approximation may affect the performance, its use is justified in the following sense: (1) evaluating the continuous policy exactly is generally intractable; (2) the approximate policy is still an exp/$q$-exp function which retains some desired properties. Developing theories in the continuous action setting is left as an interesting future work.

## 2.1 Tsallis Regularization and Sparsemax Policies

We consider a broad class of less studied entropic regularizers as $\Omega$: Tsallis entropy and Tsallis KL divergence. We can define these regularizers using $q$-logarithm in a similar manner to the standard logarithm. For $q \in \mathbb{R}_+$, we define $q$-logarithm as $\ln_q x = \frac{x^{q-1}-1}{q-1}$ and its unique inverse function $q$-exponential $\exp_q x = [1 + (q-1)x]_+^{\frac{1}{q-1}}$, where $\{\cdot\}_+ = \max\{\cdot, 0\}$, see Figure 1 for an illustration. As $q$ gets larger, $\exp_q x$ becomes more flat (second plot). Notice that $\exp_q$ is only invertible when $x > -\frac{1}{q-1}$. On the other hand, $\ln_q x$ is more peaked than the standard logarithm for input $x > 1$ and more flat otherwise (Ding & Vishwanathan, 2010). Therefore, for $0 \le \pi \le 1$, the Tsallis entropy component $-\pi \ln_q \pi$ is more flat than the Shannon entropy (third plot). Note that when $q \le 0$, Tsallis entropy becomes a convex function and Tsallis KL a concave function, therefore they are not valid regularizers (Geist et al., 2019). We consider exclusively $q > 0$ in this paper, and $q = 0$ is shown in Figure 1 only for an illustration purpose. We can define the Tsallis entropy using $q$-logarithm (Tsallis, 2009): $S_q(\pi(\cdot|s)) = -k \sum_a \pi(a|s) \ln_q \pi(a|s), k \in \mathbb{R}$. When $q \to 1$, the $q$-logarithm (resp. $q$-exponential) recovers the standard logarithm (resp. exponential) and hence Tsallis entropy degenerates to Shannon entropy. When $q = \infty$, the regularizer vanishes. When $k = \frac{1}{2}, q = 2$, we arrive at the most important non-trivial case: Tsallis sparse entropy $S_2(\pi(\cdot|s)) := \frac{1}{2} \sum_a \pi(a|s) (1 - \pi(a|s))$ (Chow et al., 2018; Lee et al., 2018). $\max_\pi \sum_a \pi(a|s) Q(s,a) + \tau S_2(\pi(\cdot|s))$ attains its maximum at $\frac{\tau}{2} \sum_{a \in K(s)} \left( \frac{Q(s,a)}{\tau} \right)^2 - \tilde{\psi} \left( \frac{Q(s,\cdot)}{\tau} \right)^2 + \frac{\tau}{2}$. The name sparse entropy comes from the fact that the regularizer leads to sparse support of the resulting *sparsemax* policy (Blondel et al., 2020; Martins & Astudillo, 2016). We compare sparsemax against two other commonly used policies argmax and softmax in Figure 2.

For $q \ne 1, 2, \infty$, the resulting policy does not have a closed-form expression. But we can apply Taylor's first-order expansion on the resulting $q$-exp function to obtain an approximate policy form: $\pi(a|s) = \exp_q \left( \frac{Q(s,a)}{\tau} - \psi \left( \frac{Q(s,\cdot)}{\tau} \right) \right)$, $\psi \left( \frac{Q(s,\cdot)}{\tau} \right) \doteq \frac{\sum_{a \in K(s)} \frac{Q(s,a)}{\tau} - 1}{|K(s)|} + \frac{q-2}{q-1}$ (Zhu et al., 2023). In Table 2 we provide a comparison on related literature to further clarify the notations used in this paper. We call this policy *sparsemax* for all $q \in \mathbb{R}_+ \setminus \{1\}$, since they truncate actions by definition of $q$-exp. $K(s)$ is the set of

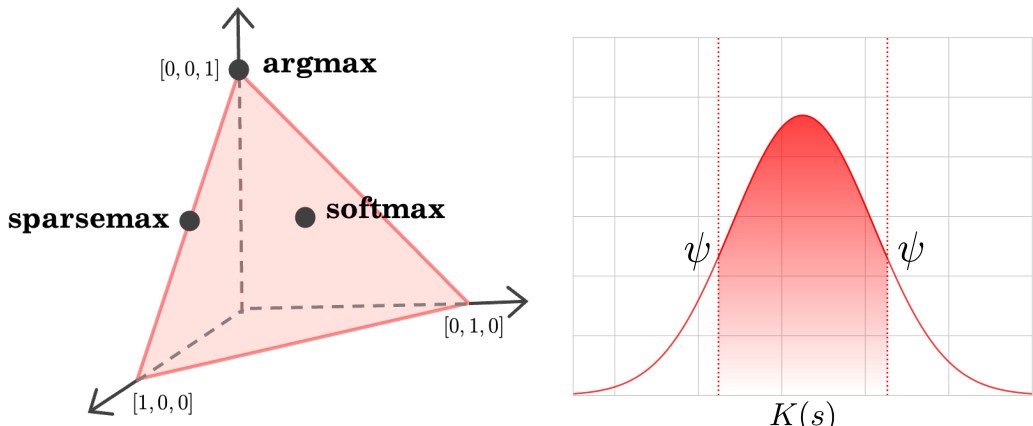

Figure 2: (Left) Comparison between argmax, softmax and sparsemax on the probability simplex. Argmax produces a deterministic policy residing on the vertices, while a softmax policy lies inside the simplex. By contrast, a sparsemax policy lives on the border. (Right) Sparsemax acting on a Gaussian policy by truncating actions with value below the threshold $\psi$. Actions with value larger than $\psi$ are collected in the set $K(s)$.

highest-value actions satisfying $1+i\frac{Q(s,a_{(i)})}{\tau} > \sum_{j=1}^{i} \frac{Q(s,a_{(j)})}{\tau}$, with $a_{(j)}$ denotes the action with $j$-th largest value. Zhu et al. (2023) recently proved that Tsallis KL divergence regularization leads to a similar policy functional $\pi(a|s) = \mu(a|s) \exp_q \left( \frac{Q(s,a)}{\tau} - \psi' \left( \frac{Q(s,\cdot)}{\tau} \right) \right)$, hence it is also a member of the sparsemax policy.

## 2.2 Offline Reinforcement Learning

We consider the problem of offline RL, where the agent cannot interact with the environment and instead learn from a fixed dataset $\mathcal{D} = \{(s,a,r,s')_{1:N}\}$ collected by some unknown behavior policy $\pi_{\mathcal{D}}$. The dataset $\mathcal{D}$ typically contains only a small subset of the $\mathcal{S} \times \mathcal{A}$ space. Standard off-policy algorithms are known to suffer from extrapolation error referring to erroneously optimistic acation values for out-of-distribution actions due to generalization capibility of function approximators. Unlike online RL, where the OOD actions can lead to more sampling around the low sample region and eventually correction of the values, in offline RL the correction is impossible since no further interaction with the environment is allowed.

We position our paper in the popular BC/imitation-based literature, where the goal of learning is to reproduce the near-expert behavior policy $\pi_{\mathcal{D}}$ (Dadashi et al., 2021; Fujimoto et al., 2019; Fujimoto & Gu, 2021; Nair et al., 2021). In this conext, explicit or implicit constraints are usually used to enforce the proximity between learned policies and $\pi_{\mathcal{D}}$ to minimize the effect of OOD actions. Explicit constraints can be implemented via density models (Ghasemipour et al., 2021; Wu et al., 2022) or in-sample constraints (Fujimoto et al., 2019; Kostrikov et al., 2022; Xiao et al., 2023) to avoid querying OOD actions. Implicit constraints are often achieved via a divergence regularizer $D$ that is added to the objective of policy improvement: $\max_\pi \mathbb{E}_{s\sim\mathcal{D}} \left[ \mathbb{E}_{a\sim\pi(\cdot|s)} [Q(s,a)] - \tau D(\pi(\cdot|s)||\pi_{\mathcal{D}}(\cdot|s)] \right)$, where $D$ is typically chosen as KL divergence. The regularization leads to the policy form $\pi(a|s) \propto \pi_{\mathcal{D}}(a|s) \exp \left( \tau^{-1} Q(s,a) \right)$ where the learned policy conditions on the support of the behavior policy and weighted by exponential of action values (advantage) (Peng et al., 2020; Siegel et al., 2020; Nair et al., 2021). However, as shown by (Rudner et al., 2021), KL regularization can lead to pathologies of the learned policies such as vanishing variances and exploding gradients. In this paper, we propose to use the Tsallis KL regularization which generalizes KL divergence and offers more flexibility over the standard KL choice (e.g., AWAC). However, investigation of pathologies is beyond the scope and we leave it to future work.

# 3  Matching In-sample Constraint By Sparsemax Truncation

To alleviate the OOD error, Fujimoto et al. (2019) proposed the in-sample Bellman optimality equation that modifies the Q-learning update target to only for the actions present in the dataset:

$$Q_{*,\pi_{\mathcal{D}}}(s,a) = r(s,a) + \gamma \mathbb{E}_{s' \sim P(\cdot|s,a)} \left[ \max_{a':\pi_{\mathcal{D}}(a'|s')>0} Q_{*,\pi_{\mathcal{D}}}(s',a') \right]. \tag{1}$$

This scheme was extended to the distributional RL setting by Kostrikov et al. (2022). Recently, Xiao et al. (2023) proposed the in-sample softmax Bellman optimality equation:

$$Q_{*,\pi_{\mathcal{D}}}(s,a) = r(s,a) + \gamma \mathbb{E}_{s' \sim P(\cdot|s,a)} \left[ \tau \ln \sum_{a':\pi_{\mathcal{D}}(a'|s')>0} \exp \left( \tau^{-1} Q_{*,\pi_{\mathcal{D}}}(s',a') \right) \right]. \tag{2}$$

In-sample softmax policy takes the form $\pi_{*,\pi_{\mathcal{D}}}^{\texttt{softmax}}(a|s) \propto \pi_{\mathcal{D}}(a|s) \exp \left( \frac{Q_{*,\pi_{\mathcal{D}}}(s,a)}{\tau} - \ln \pi_{\mathcal{D}}(a|s) \right)$, the subtracted term $\ln \pi_{\mathcal{D}}$ is to make sure not tightly follow the behavior policy when it is not good.

Let us consider regularization with $\tau S_q(\pi(\cdot|s))$. This change of regularizer embodies a fundamentally different assumption: we can link the fact that the dataset contains only a subset of actions to sparsemax behavior policies. Indeed, if we assume the dataset was generated by the behavior policy in a manner such that an action is sampled with probability proportional to its value (Kostrikov et al., 2021), then we can model the behavior policy by a sparsemax policy. This assumption is mild since softmax policy is a member of sparsemax when $q = 1$. To formalize the idea, let us assume the behavior policy is within the sparsemax policy class induced by Tsallis entropy $\pi_{\mathcal{D}}(a|s) \propto \exp_q \left( \frac{Q_{\pi_{\mathcal{D}}}(s,a)}{\tau_{\mathcal{D}}} \right)$, $\sum_{a \in K_{\mathcal{D}}(s)} \pi_{\mathcal{D}}(a|s) = 1$, where $\tau_{\mathcal{D}}$ is an unknown coefficient and $K_{\mathcal{D}}(s)$ denotes the set of actions present in the dataset. Therefore, we can replace the in-sample constraint $a : \pi_{\mathcal{D}}(a|s) > 0$ to the truncation criterion $a \in K_{\mathcal{D}}(s)$:

$$Q_{t+1,\pi_{\mathcal{D}}}(s,a) = r(s,a) + \gamma \mathbb{E}_{s' \sim P(\cdot|s,a)} \left[ \max_{\pi} \sum_{a' \in K_{\mathcal{D}}(s)} \pi(a'|s') \left( Q_{t,\pi_{\mathcal{D}}}(s',a') + \tau S_q(\pi(\cdot|s')) \right) \right].$$
$$\pi_{t+1,\pi_{\mathcal{D}}}(a|s) \propto \exp_q \left( \frac{Q_{t,\pi_{\mathcal{D}}}(s,a)}{\tau} \right), \quad \sum_{a \in K_{\mathcal{D}}(s)} \pi_{t+1,\pi_{\mathcal{D}}}(a|s) = 1. \tag{3}$$

This replacement hints at two potential benefits of applying in-sample Tsallis regularization: (i) suppose the action values are fixed, then a sparsemax policy could extract a new subset of allowable actions from $K_{\mathcal{D}}$, and this procedure could continue until there is only the highest-valued action in the set. (ii) Sparsemax policies generated recursively from (i) may have finite KL divergence, which in turn implies that the KL between the $\pi_t$ and $\pi_{\mathcal{D}}$ may be bounded. We discuss (i) here and leave (ii) to next section.

**Sparsemax interpolates softmax and hard-max.**  Continuing the discussion of Eq. (3), let us define $K_{0,q}(s) := K_{\mathcal{D}}(s)$ as our starting point. Then during learning, a new subset of allowable actions $K_{t,q}(s)$ depending on $q$ and iteration $t$ can be extracted from $K_{0,q}(s)$. Recursively, every $K_{t+1,q}$ contains only a subset of actions from the last set $K_{t,q}$, or mathematically speaking, $K_{0,q} \succeq K_{1,q} \succeq K_{2,q} \succeq \cdots \succeq K_{t,q} \succeq K_{t+1,q}$, where we used $A \succeq B$ to denote $A$ is a subset of $B$. To be more concrete, let us consider $q = 2$ where the $q$-maximum has an analytic solution. From Section 2.1 it is clear that:

$$Q_{t+1,\pi_{\mathcal{D}}}(s,a) = r(s,a) + \gamma \mathbb{E}_{s' \sim P(\cdot|s,a)} \left[ \frac{\tau}{2} \sum_{a' \in K_{t,2}(s)} \left( \frac{Q_{t,\pi_{\mathcal{D}}}(s',a')}{\tau} \right)^2 - \tilde{\psi} \left( \frac{Q_{t,\pi_{\mathcal{D}}}(s',\cdot)}{\tau} \right)^2 + \frac{\tau}{2} \right]. \tag{4}$$

Notice the constraint under the summation was replaced from $K_{\mathcal{D}}(s)$ in Eq. (3) to $a' \in K_{t,2}(s)$, due to the aforementioned recursive extraction. However, it is worth noting that the recursive extraction may not hold true as action values can change during updates.

The above scheme regularizes both policy improvement and policy evaluation. Another possibility is to regularize only policy improvement by choosing Tsallis KL divergence as the regularizer. We use the following policy iteration schemes to help clarify the difference:

$$
(a) \begin{cases} \pi_{t+1} = \arg\max_\pi \mathbb{E}_\pi \left[ Q_t(s,a) + \tau \ln_q \pi(a|s) \right] \\ Q_{t+1} = r(s,a) + \gamma \mathbb{E}_{P,\pi} \left[ Q_t(s',a') + \tau \ln_q \pi_{t+1}(a'|s') \right], \end{cases} \quad (b) \begin{cases} \pi_{t+1} = \arg\max_\pi \mathbb{E}_\pi \left[ Q_t(s,a) - \tau \ln_q \frac{\pi(a|s)}{\pi_t(a|s)} \right] \\ Q_{t+1} = r(s,a) + \gamma \mathbb{E}_{P,\pi} \left[ Q_t(s',a') \right], \end{cases}
$$
(5)

where (a) corresponds to **Tsallis entropy + regularize both** and (b) **Tsallis KL + policy improvement only**. It is worth noting that both schemes fall into the mirror descent policy iteration framework (Vieillard et al., 2020, Eq.(1)). In the offline RL context, the first scheme (regularizing both policy improvement and evaluation) is discussed in (Wu et al., 2020); and the second scheme is the basis of advantage weighted actor-critic (AWAC) (Nair et al., 2021). We present in Section 5 implementation for both schemes. Tsallis KL is chosen for the second scheme with the following reason: we notice that augmenting the value function with divergence often results in unstable learning possibly due to the existence of policy ratios.

## 4 In-sample Sparsemax Policies Have $q$-bounded KL Divergence

A sparsemax policy can be expressed as a $q$-exponential policy. Therefore, existing theoretical results on $q$-statistics (Yamano, 2002; Ding & Vishwanathan, 2010) may provide a clue to characterizing the similarity between two sparsemax policies. We are especially interested in characterizing the KL divergence between a learned policy and the behavior policy, which is particularly important for BC/imitation-based methods (Fujimoto et al., 2019; Fujimoto & Gu, 2021; Wu et al., 2020; 2022). However, bounding potentially unbounded KL divergence is in general a difficult task, if not impossible (McAllester & Stratos, 2020). Instead, we do not seek to provide a general upper bound but rather aim at quantifying the dependency of KL divergence on the intermediate variable $q$.

The following theorem states that the upper bound on KL divergence between learned/behavior policies can be flexibly adjusted by changing $q$. *Therefore, our prior knowledge regarding the quality of the dataset may impact our choice of $q$*: when we are confident that sufficient near-expert trajectories exist, it may be preferable to choose a large $q$ to encourage more similarity to $\pi_\mathcal{D}$; on the other hand, choosing $q = 1$ may be more robust to a non-expert dataset.

**Theorem 1.** *Suppose the dataset $\mathcal{D}$ is generated by sparsemax behavior policies such that for each state there is at most one behavior policy of arbitrary entropic index $q$. Let $K_{t,q}(s) \preceq K_\mathcal{D}(s)$ denote the set of allowable actions of the learned sparsemax policy at $t$-th iteration. Then the following $q$-conditional upper bound of KL divergence can be derived for sparsemax policies $\pi_t$ and $\pi_\mathcal{D}$:*

$$
D_{KL}(\pi_t(\cdot|s) \,||\, \pi_\mathcal{D}(\cdot|s)) \leq \begin{cases} |K_{t,q}(s)| \, \Pi_{\mathcal{D},t,q}, & \text{if } \pi_t \preceq \pi_\mathcal{D}, \ 1 < q < \infty \\ \infty, & \text{otherwise} \end{cases}
$$
(6)

*where $\Pi_{\mathcal{D},t,q} := \pi_t^q(a|s) - \pi_t^{q-1}(a|s) + \pi_t(a|s) - \frac{q-3}{q-1} + \pi_\mathcal{D}^{q-2}(a|s) - \pi_\mathcal{D}^{q-1}(a|s)$.*

*Proof.* See Appendix B for the proof. □

We may interpret the term $\pi_t(a|s)$ as a baseline having a fixed power $\pi_t^{q=1}(a|s)$. When $q = 1$, the upper bound may be $\infty$. On the other hand, when $q \to \infty$, the learned policy $\pi_t$ approaches the $\arg\max$ (Lee et al., 2020), and the upper bound approaches zero. Intuitively, when $q = \infty$, $K_{t,q}(s)$ has only one action, which corresponds to assuming the behavior policy is an $\arg\max$ and it is identified by $\pi_t(a|s) = 1$, therefore the KL is zero when the their supports agree, or $\infty$ otherwise.

We can replace the reference policy $\pi_\mathcal{D}$ in equation 6 to a learned policy $\pi_{t-1}$ if the assumption of fixed action values holds, since $\pi_t \preceq \pi_{t-1}$. Therefore, the bound provides a means to understanding the distance between sparsemax policies. For $q = 1$ (the in-sample softmax case) the bound is not useful and simply states the KL divergence may be unbounded. On the other hand, choosing any $q > 1$ brings an upper

bound of at most $4|K_{t,q}(s)|$. When $q = 2$, the in-sample sparsemax has KL divergence to the behavior policy bounded by $|K_{t,q}(s)| (\pi_t(a|s) - \pi_{\mathcal{D}}(a|s) + 2)$. In general, there is a trade-off between $\pi^q$ and $|K_{t,q}(s)|$: $K_{t,q}(s)$ tends to collect less actions when $q \to \infty$. Again, we should note that the theorem holds only when $K_{\mathcal{D}} \succeq K_{1,q} \succeq K_{2,q} \succeq \cdots \succeq K_{t,q}$, as per the discussion for equation 4. It should also be noted that the bound is only applicable to discrete action spaces. Moreover, staying close to the behavior policy may not always be preferable, especially when the dataset contains too many poor trajectories.

## 5 In-sample Sparsemax Actor-Critic

We propose Tsallis Advantage Weighted Actor-Critic (Tsallis AWAC) based on the Tsallis KL divergence regularization. We also discuss another new algorithm Tsallis In-Sample Actor-Critic (Tsallis InAC) based on Tsallis entropy. However, as will be analyzed shortly, Tsallis InAC may not necessarily introduce sparsity and does not enjoy straightforward policy evaluation. Nonetheless, we include it in analysis and experiments for completeness.

To fulfill the in-sample constraint we leverage only those actions present in the offline dataset but not sampled actions to learn the actor. The architecture is similar to soft actor-critic (Haarnoja et al., 2018): let $\theta$ denote the parametrization of the critic $Q_\theta$, $\phi$ the actor $\pi_\phi$ and $\zeta$ the state value function $V_\zeta$. Another network $\omega$ is trained by maximum likelihood $-\mathbb{E}_{(s,a)\sim\mathcal{D}} [\log \pi_\omega(a|s)]$ to imitate the behavior policy $\pi_\omega \approx \pi_{\mathcal{D}}$.

Before deriving the Tsallis AWAC loss functions, we recall the actor loss for AWAC:

$$\mathcal{L}_{\text{actor}}^{\texttt{AWAC}}(\phi) = -\mathbb{E}_{(s,a)\sim\mathcal{D}} \left[ \exp\left( \frac{Q_\theta(s,a) - V_\zeta(s)}{\tau} \right) \ln \pi_\phi(a|s) \right], \tag{7}$$

which is a common actor loss function for advantage-weighted regression methods (Wang et al., 2020; Siegel et al., 2020; Nair et al., 2021; Xu et al., 2023; Garg et al., 2023). It can also be derived from the $(b)$ scheme in Eq. (5) by replacing Tsallis KL to standard KL. Intuitively, minimizing $\mathcal{L}_{\text{actor}}^{\texttt{AWAC}}(\phi)$ results in maximizing the log-likelihood of actions in the dataset $a \sim \mathcal{D}$ and implicitly minimizing the likelihood for these not in the dataset, as the probabilities of all actions should sum to one. The degree of maximization is controlled by the exponential advantage of an action $\exp\left( \frac{Q_\theta(s,a) - V_\zeta(s)}{\tau} \right)$: when an action has high $Q_\theta(s,a) - V_\zeta(s)$, the policy gets updated to increase its probability more, which may be problematic when the value estimates are poor. By contrast, in-sample softmax (Xiao et al., 2023) compensates for this fact by subtracting a $\ln \pi_{\mathcal{D}}$ term inside the exponential. For the proposed Tsallis methods, a crucial difference lies in how this exponential advantage function is modified.

**Tsallis AWAC (TAWAC).** This scheme regularizes only the policy improvement step as can be seen from Eq. (5). Given $Q_\theta(s,a)$, we write the policy as $\hat{\pi}_{Q_\theta,\pi_{\mathcal{D}}}^{\texttt{TAWAC}}(a|s) \propto \pi_{\mathcal{D}}(a|s) \exp_q\left( \frac{1}{\tau} Q_\theta(s,a) \right)$. Repeating the KL loss step, we have:

$$D_{KL}\left( \hat{\pi}_{Q_\theta,\pi_{\mathcal{D}}}^{\texttt{TAWAC}}(\cdot|s) \,||\, \pi_\phi(\cdot|s) \right) = \mathbb{E}_{a\sim\hat{\pi}_{Q_\theta,\pi_{\mathcal{D}}}^{\texttt{TAWAC}}(\cdot|s)} \left[ \ln \hat{\pi}_{Q_\theta,\pi_{\mathcal{D}}}^{\texttt{TAWAC}}(a|s) - \ln \pi_\phi(a|s) \right]$$

$$= \mathbb{E}_{a\sim\pi_{\mathcal{D}}} \left[ \exp_q\left( \frac{Q_\theta(s,a)}{\tau} - \psi\left( \frac{Q_\theta(s,\cdot)}{\tau} \right) \right) \left( \ln \hat{\pi}_{Q_\theta,\pi_{\mathcal{D}}}^{\texttt{TAWAC}}(a|s) - \ln \pi_\phi(a|s) \right) \right].$$

The normalization function $\psi$ poses a challenge to continuous-action problems. Inspired by (Xiao et al., 2023), we propose to replace it with the state value function $V_\zeta$. We write the loss functions as:

$$\mathcal{L}_{\text{actor}}^{\texttt{TAWAC}}(\phi) = -\mathbb{E}_{(s,a)\sim\mathcal{D}} \left[ \exp_q\left( \frac{Q_\theta(s,a) - V_\zeta(s)}{\tau} \right) \ln \pi_\phi(a|s) \right],$$

$$\mathcal{L}_{\text{critic}}^{\texttt{TAWAC}}(\theta) = \mathbb{E}_{(s,a,r,s')\sim\mathcal{D}} \left[ \left( r + \gamma V_\zeta(s') - Q_\theta(s,a) \right)^2 \right], \tag{8}$$

$$\mathcal{L}_{\text{baseline}}^{\texttt{TAWAC}}(\zeta) = \mathbb{E}_{s\sim\mathcal{D}, a\sim\pi_\phi(\cdot|s)} \left[ \left( V_\zeta(s) - Q_\theta(s,a) \right)^2 \right].$$

**Remark.** Comparing $\mathcal{L}_{\text{actor}}^{\texttt{TAWAC}}(\phi)$ and $\mathcal{L}_{\text{actor}}^{\texttt{AWAC}}(\phi)$, it is clear that even though Tsallis AWAC is only different to AWAC in the $q$-exponential function, it brings sparsity to the policy. To see this, we can write out $\mathcal{L}_{\text{actor}}^{\texttt{TAWAC}}(\phi)$

as the following:

$$-\mathbb{E}_{(s,a)\sim\mathcal{D}}\left[\mathbb{1}\left\{\left[1+(q-1)\left(\frac{Q_\theta(s,a)-V_\zeta}{\tau}\right)\right]^{\frac{1}{q-1}}>0\right\}\cdot\left(\left[1+(q-1)\left(\frac{Q_\theta(s,a)-V_\zeta(s)}{\tau}\right)\right]^{\frac{1}{q-1}}\right)\ln\pi_\phi(a|s)\right].$$

Since the root does not affect the sign, it can be seen that actions with values $\frac{Q_\theta(s,a)-V_\zeta(s)}{\tau}<-\frac{1}{q-1}$ will be truncated: the actor loss for these actions becomes zero and TAWAC does not maximize its likelihood. Another interesting observation is that by setting $q=2$, $\mathcal{L}_{\text{actor}}^{\texttt{TAWAC}}(\phi)$ **recovers the sparse $Q$-learning objective in** (Xu et al., 2023), which was derived from the $\alpha$-divergence perspective. However, our loss is more general since for all $q>0$ $\mathcal{L}_{\text{actor}}^{\texttt{TAWAC}}(\phi)$ has a truncation effect. By contrast, all exponential advantage weighted methods do not have sparsity and can maximize likelihood.

**Tsallis In-Sample Actor-Critic (Tsallis InAC).** Different from Tsallis AWAC that has a clean policy form, Tsallis entropy regularization induces a much more complicated policy (see Appendix D):

$$\hat{\pi}_{Q_\theta,\pi_\mathcal{D}}^{\texttt{TInAC}}(a|s)=\pi_\mathcal{D}(a|s)\left(\exp_q\left(\frac{1}{\tau}Q_\theta(s,a)+\ln_q\frac{1}{\pi_\mathcal{D}(a|s)}\right)^{q-1}+(q-1)^2\frac{1}{\tau}Q_\theta(s,a)\ln_q\frac{1}{\pi_\mathcal{D}(a|s)}\right)^{\frac{1}{q-1}}.$$

It can be seen that **the Tsallis InAC policy cannot contain everything inside the $q$-exp function**. This can lead to numerical issues and more importantly, the loss of sparsity. We found that Tsallis InAC in general underperforms Tsallis AWAC. Two potential reasons are identified. Firstly, the Tsallis InAC actor loss in general does not lead to sparsity, since there is a $\frac{Q_\theta(s,a)\ln_2\frac{1}{\pi_\omega(a|s)}}{\tau}$ term outside the $q$-exponential function. This term is very sensitive to the sign of $Q$ and magnifies its value by the term $\ln_2\frac{1}{\pi}$ which is typically large. Directly from this, the second point is that Tsallis InAC may be drastic in deciding good/bad actions simply by its sign and can be minimizing likelihood of bad actions. This stands in sheer contrast to InAC and Tsallis AWAC, which do not explicitly minimize likelihood. The drastic behavior of Tsallis InAC in general leads to undesired performance, but with exception for expert datasets, see Appendix D.

## 6 Experiments

Below, we evaluate Tsallis AWAC against several offline reinforcement learning methods. The goal of the experiments is to find when is teh proposed method best applied in offline RL, as well as gain insight into the effect of the $\tau$ and $q$ parameters (especially as we enter regimes where there aren't closed form expressions for the policies). Finally, we evaluate if the upper bound holds in the continuous action setting.

**Domain Details.** We compare the proposed method against a number of existing algorithms on standard benchmark D4RL environments (Fu et al., 2020). Specifically, we use three datasets from the Mujoco suite in D4RL. Trajectories in the offline datasets are collected by a SAC agent. The naming expert/medium expert/medium replay reflects the level of the trained agent used to collect the transitions. The expert dataset contains trajectories collected by a fully trained SAC agent; the medium dataset contains 1M samples generated from an partially-trained SAC policy. Medium-expert combines the trajectories of the expert and the medium. Medium-replay consists of samples in the replay buffer during training until the policy reaches the medium level of performance.

**Compared algorithms.** We compare against a number of existing algorithms: InAC: in-sample softmax actor-critic (Xiao et al., 2023). It is worth noting InAC can be seen as the special case $q=1$ of Tsallis InAC. TD3 + BC (Fujimoto & Gu, 2021) augments the policy improvement step of TD3 with an additional behavior cloning (BC) term $(\pi(s)-a)^2$ as indicated by (Xiao et al., 2023). This term can be seen as a KL regularization under the Gaussian policy parametrization. IQL: implicit Q-learning (Kostrikov et al., 2022) employed in-sample hard max Eq. (1). AWAC: advantage-weighted actor-critic (Nair et al., 2021), it can be as a special case of Tsallis AWAC when $q=1$. For all the compared algorithms except InAC, we followed the published settings of the compared agents. Implementation details can be found in Appendix C.

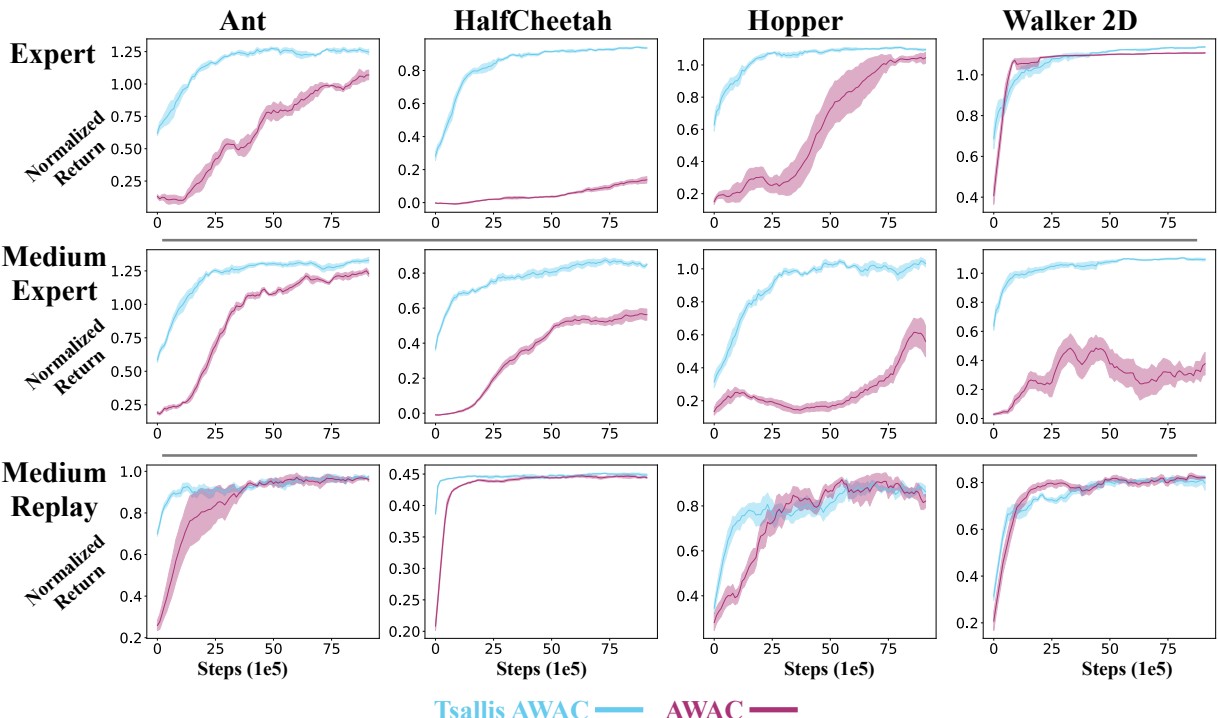

Figure 3: Comparison between Tsallis AWAC and AWAC. Normalized policy evaluation scores on Mujoco Medium Replay dataset over one million steps. Results are the average over 5 runs with ribbon denoting the standard error. $y$-axis: scores; $x$-axis: number of iterations.

| Dataset | BC | UAWAC | AWAC | TD3BC | IQL | InAC | Tsallis InAC | Tsallis AWAC |
|---|---|---|---|---|---|---|---|---|
| walker2d-E | $106.7 \pm 0.2$ | $108.4 \pm 0.4$ | $110.2 \pm 0.03$ | $79.4 \pm 1.0$ | $102.9 \pm 0.6$ | $110.6 \pm 0.1$ | $108.7 \pm 0.00$ | $\mathbf{113.2 \pm 0.00}$ |
| walker2d-ME | $90.1 \pm 13.2$ | $96.5 \pm 9.1$ | $51.2 \pm 1.3$ | $89.7 \pm 0.6$ | $96.0 \pm 0.4$ | $\mathbf{109.0 \pm 0.1}$ | $0.85 \pm 0.02$ | $101.1 \pm 0.02$ |
| walker2d-MR | $20.3 \pm 9.8$ | $23.6 \pm 6.9$ | $77.4 \pm 0.3$ | $17.2 \pm 0.4$ | $66.3 \pm 0.5$ | $69.8 \pm 0.6$ | $0.74 \pm 0.01$ | $\mathbf{80.4 \pm 0.01}$ |
| halfcheetah-E | $91.8 \pm 1.5$ | $92.9 \pm 0.6$ | $10.0 \pm 0.1$ | $92.1 \pm 0.03$ | $91.5 \pm 0.2$ | $\mathbf{93.6 \pm 0.04}$ | $93.1 \pm 0.00$ | $92.7 \pm 0.00$ |
| halfcheetah-ME | $44.0 \pm 1.6$ | $42.7 \pm 0.3$ | $38.5 \pm 0.3$ | $54.1 \pm 0.4$ | $83.4 \pm 0.5$ | $83.5 \pm 0.4$ | $50.2 \pm 0.01$ | $\mathbf{84.5 \pm 0.01}$ |
| halfcheetah-MR | $37.6 \pm 2.1$ | $35.9 \pm 3.7$ | $44.8 \pm 0.01$ | $35.0 \pm 0.2$ | $45.0 \pm 0.03$ | $44.3 \pm 0.02$ | $37.9 \pm 0.01$ | $\mathbf{44.9 \pm 0.00}$ |
| hopper-E | $107.7 \pm 0.7$ | $\mathbf{110.5 \pm 0.5}$ | $45.0 \pm 0.1$ | $78.9 \pm 0.9$ | $89.4 \pm 0.7$ | $103.4 \pm 0.4$ | $110.3 \pm 0.00$ | $109.9 \pm 0.00$ |
| hopper-ME | $53.9 \pm 4.7$ | $44.9 \pm 8.1$ | $23.3 \pm 0.4$ | $50.8 \pm 0.3$ | $61.8 \pm 1.0$ | $93.8 \pm 0.7$ | $63.4 \pm 0.03$ | $\mathbf{101.2 \pm 0.02}$ |
| hopper-MR | $16.6 \pm 4.8$ | $25.3 \pm 1.7$ | $71.2 \pm 0.1$ | $20.3 \pm 0.2$ | $60.3 \pm 0.5$ | $92.1 \pm 0.4$ | $26.7 \pm 0.00$ | $\mathbf{86.4 \pm 0.02}$ |
| ant-E | - | - | $100.6 \pm 0.9$ | $93.3 \pm 1.0$ | $118.8 \pm 0.5$ | $\mathbf{128.4 \pm 0.4}$ | $125.4 \pm 0.01$ | $125.2 \pm 0.01$ |
| ant-ME | - | - | $97.8 \pm 1.6$ | $88.7 \pm 0.9$ | $121.0 \pm 0.6$ | $120.9 \pm 0.6$ | $114.2 \pm 0.00$ | $\mathbf{130.2 \pm 0.01}$ |
| ant-MR | - | - | $65.0 \pm 1.2$ | $59.2 \pm 0.4$ | $89.3 \pm 0.4$ | $88.4 \pm 0.6$ | $68.7 \pm 0.00$ | $\mathbf{96.0 \pm 0.01}$ |

Table 1: Final performance of compared algorithms, E: expert; ME: medium expert; MR: medium replay. Scores are reproduced from (Xiao et al., 2023) and (Lyu et al., 2022). Tsallis AWAC and InAC perform favorably across various datasets, while Tsallis InAC is mostly competitive in expert-level datasets only. Notice that both Tsallis InAC and Tsallis AWAC tend to be stable by having negligible standard deviation.

## 6.1 Comparison Against the Existing Methods

Since Tsallis AWAC generalizes AWAC, we first compare the generalizations in Figure 3, and leave the comparison against all methods to Figure 4 and Table 1.

**Tsallis AWAC against AWAC.** From Figure 3 it is visible Tsallis AWAC outperforms AWAC by a large margin on Expert and Medium-Expert datasets and the performance remains stable across all datasets. The poor performance of AWAC has been discussed a lot in the literature (Xiao et al., 2023; Xu et al., 2023): the exponential term in $\mathcal{L}_{\text{actor}}^{\text{AWAC}}$ can cause unstable gradients and is also more vulnerable to hyperparameter choices. On the other hand, the favorable performance of Tsallis AWAC confirms the discussion given by

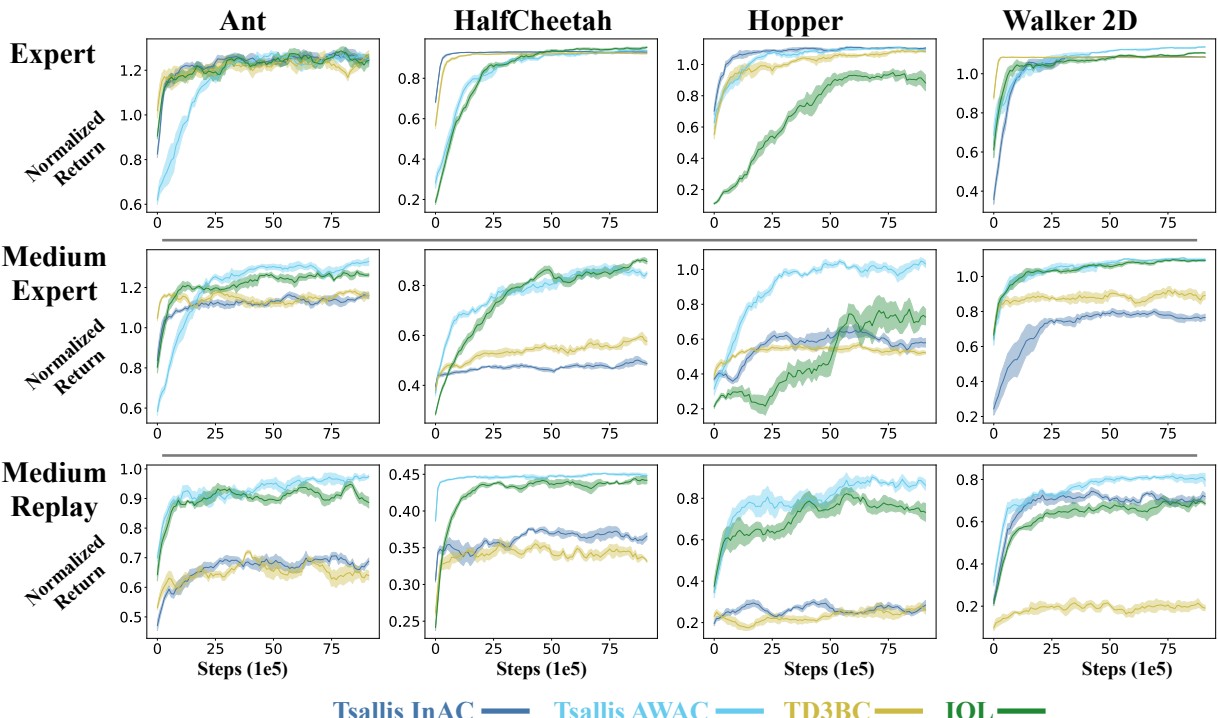

Figure 4: Additional comparison against TD3BC and Implicit Q-learning. Normalized policy evaluation scores over one million steps and are the average over 5 runs, with ribbon denoting the standard error. $y$-axis: scores; $x$-axis: number of iterations.

the remark after equation 7: the sparsity-inducing learning objective $\mathcal{L}_{\text{actor}}^{\texttt{TAWAC}}(\phi)$ is more robust against both transition/suboptimality noises and numerical errors than the exponential function (Xu et al., 2023).

**Against all methods.** From Figure 4 it is visible that Tsallis InAC and TD3BC are the best performers on expert level datasets—in terms of convergence speed and the final score. By the analysis for Tsallis InAC, the term $\ln_q \frac{1}{\pi_{\mathcal{D}}}$ may dominate the actor and render Tsallis InAC behave like a BC method even without an explicit BC term. This is especially apparent on `HalfCheetah-expert` and `Hopper-expert` where Tsallis InAC learned even faster than TD3BC. Both TD3BC and Tsallis InAC drastically degrade for Medium Expert and Medium Replay datasets. Tsallis AWAC outperforms IQL on almost all non-expert datasets, which matches the observation of (Xiao et al., 2023): IQL may perform poorly when the data distribution is skewed towards suboptimal actions in some states, pulling down the expectile regression targets.

To confirm our conclusion, we show in Table 1 the final performance of all the compared algorithms. We also add behavior cloning (BC) and Uncertainty AWAC (UAWAC) (Wu et al., 2021) for further comparison. Scores of the BC and UAWAC are quoted from (Lyu et al., 2022). It is visible that Tsallis AWAC and InAC are the top performers across datasets, with Tsallis AWAC achieving the best scores on 8 out of 12 datasets. This result is encouraging and testifies to the power of sparsity brought by the $q$-exp formulation, since Tsallis AWAC differs from other AWAC-based methods mostly in the exponential function. By contrast, Tsallis InAC is only competitive on expert-level datasets. Upon examining Table 1 it is visible that in medium datasets BC, TD3BC and Tsallis InAC perform poorly, which echoes the findings in (Kumar et al., 2022). On the other hand, Tsallis AWAC performs stably across datasets thanks to the introduced sparsity. Investigating Tsallis AWAC beyond the current setting is therefore an interesting future direction.

## 6.2 The Importance of $q$ and $\tau$

Entropic index $q$ and regularization coefficient $\tau$ determine the degree of sparsity. Therefore, they are related to performance and the $q$-conditional distance to the behavior policy as shown by equation 6. We evaluate

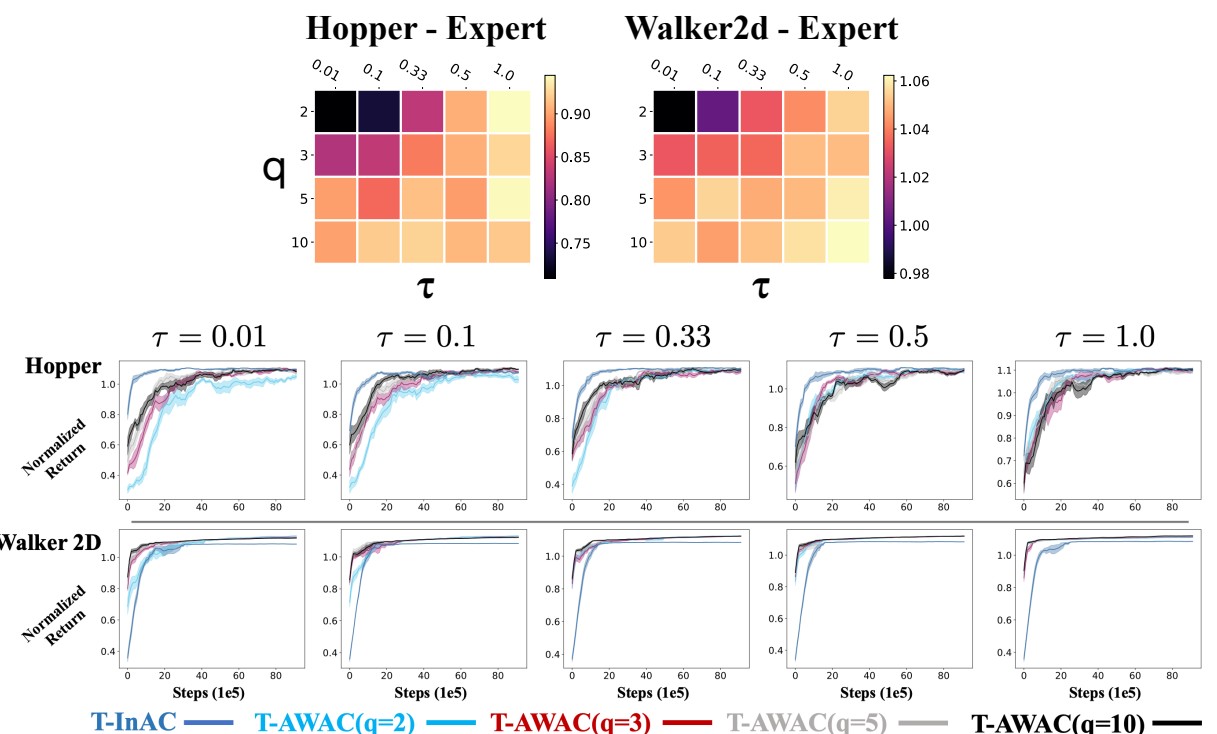

Figure 5: (Upper) Heatmap of average normalized score of Tsallis AWAC with different $q$, evaluated over first 500K steps of training. (Lower) Normalized score learning curves of Tsallis AWAC with different $q$.

different $(q, \tau)$ combinations of Tsallis AWAC in Figure 5. Performance is evaluated after $5 \times 10^5$ steps of training. It is visible from the color maps that for the first half of learning, Tsallis AWAC on both environments exhibited preference to larger $q$: they tend to learn quicker and are relatively insensitive to $\tau$; while smaller $q$ such as $q = 2$ prefers larger $\tau$. Consider fixed $\tau$, by definition of the Tsallis AWAC policy $\pi_{\mathcal{D}}(a|s) \exp_q \left( \frac{Q_\theta(s,a) - V_\zeta(s)}{\tau} \right) = \pi_{\mathcal{D}}(a|s) \left[ 1 + (q-1) \frac{Q_\theta(s,a) - V_\zeta(s)}{\tau} \right]_+^{\frac{1}{q-1}}$, the difference $Q_\theta(s,a) - V_\zeta(s)$ becomes less significant with under power function with larger power $\frac{1}{q-1}$ (imagine softmax with a large $\tau$) and the truncation threshold $-\frac{1}{q-1}$. Comparing to $q = 2$ where $\exp_q$ is linear in its argument, higher $q$ truncates more, especially for actions that $Q_\theta(s,a) - V_\zeta(s) < 0$. Let us in turn consider fixed $q$. Larger $\tau$ results in $|K_{t,q}|$. Similar to the softmax case (Haarnoja et al., 2017), it is advocated that choosing a larger $\tau$ at the beginning and gradually annealed towards zero. Therefore, when the dataset contains sufficient (near) expert trajectories, choosing a large $(q, \tau)$ may accelerate learning at the early stage. In Figure 6 it is visible that on the expert level environments all Tsallis agents seem to be robust and converged to policies of similar level.

## 6.3   KL divergence to the Behavior Policy

Though higher similarity to the behavior policy does not necessarily imply better performance, we quantitatively evaluate it in support of our theory Eq. (6). We plot in Figure 7 $D_{KL}(\pi_\omega(\cdot|s) \,||\, \pi_\phi(\cdot|s))$ for Tsallis AWAC with different $q$ and Tsallis InAC throughout training. Since the KL divergence can be written as $\mathbb{E}_{a \sim \pi_\omega(\cdot|s)} [\ln \pi_\omega(a|s) - \ln \pi_\phi(a|s)]$, and $\pi_\omega$ is imitating $\pi_{\mathcal{D}}$, we replace the sampling part to $a \sim \pi_{\mathcal{D}}(\cdot|s)$ to allow for random sampling actions from the dataset to compute the log-policy difference. Though in the continuous action setting $K_{t,q}$ is uncountable and the upper bound on KL is no longer valid, we can empirically investigate it. On Hopper-expert Tsallis AWAC for all $q$ converged to 0, and Tsallis InAC remains stable around 1; different $q$ did not seem to affect the KL divergence. On the other hand, on Walker2d-expert a clear stratification was displayed: larger $q$ resulted in lower divergence value. Theoretically, the RHS of

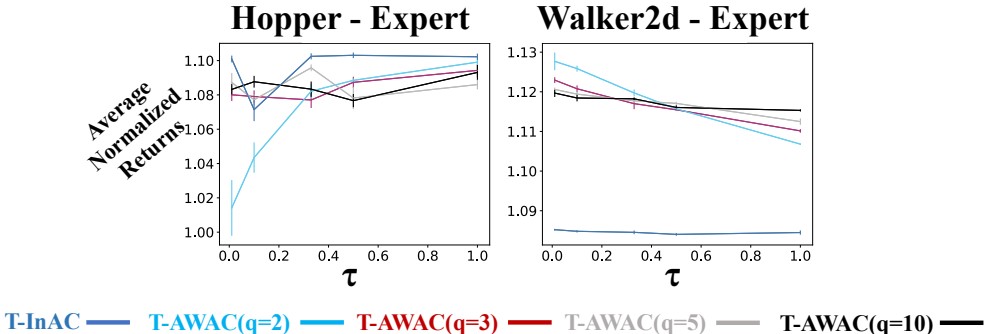

Figure 6: Sensitivity of Tsallis AWAC to $\tau$. Normalized scores averaged over the final 500k steps of training.

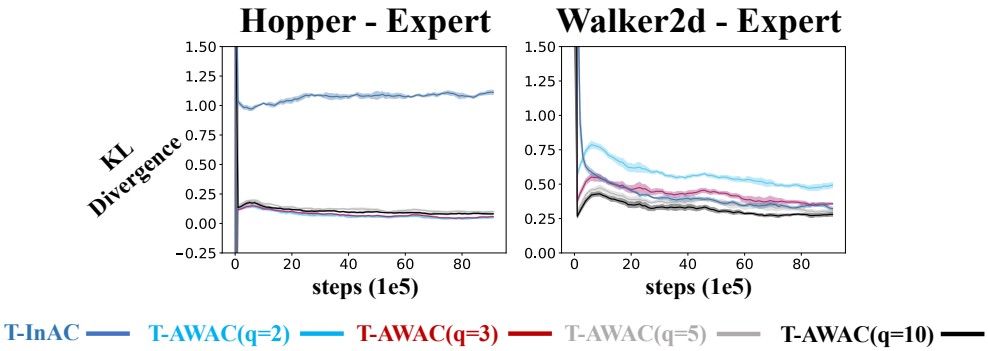

Figure 7: $D_{KL}(\pi_\omega(\cdot|s) \,||\, \pi_\phi(\cdot|s))$ throughout training for different $q$ with best $\tau$.

Eq. (6) approaches $|K_{t,q}(s)| \left(\pi_t(a|s) - \frac{q-3}{q-1}\right)$ as $q$ gets larger, which is indeed tighter than $q = 2$. Therefore, the upperbound may still be in effect even in the continuous action setting.

## 7 Limitations of The Proposed Method

We recognize that the paper has some limitations. Firstly, the upper bound in Theorem 1 was only $q$-conditional, and it necessitated the assumption that the ordering between action values remained unchanged. Deriving a generally applicable upper bound on KL divergence is difficult, if not impossible (McAllester & Stratos, 2020). Utilizing other tools instead of $q$-statistics for derivation is an interesting future direction.

Secondly, as seen in Section 5, Tsallis InAC induced by Tsallis entropy had a very complicated policy form that failed to contain everything inside the $q$-exp function. As a result, Tsallis InAC could have numerical issues and lose the sparsity by $q$-exp, as compared to Tsallis AWAC. The experimental results confirmed that Tsallis InAC indeed performed poorly especially on non-expert datasets. How to leverage $q$-statistics literature to address these problems remains an open question.

## 8 Discussion and Conclusion

Tsallis regularizers have been less popular in RL due to the action truncation of its regularized optimal policy. The truncation often leads to underperformance in online problems resulting from limited exploration. This paper is the first work that introduces Tsallis regularizers to offline RL, where no exploration is required. Tsallis regularizers bring close two popular offline RL methods avoiding producing erronesouly optimistic out-of-distribution actions: divergence regularization and in-sample constraint. Tsallis regularization induces sparsemax policies that truncate actions with low action values, which we exploited to link to the fact that

offline datasets contain only a subset of actions: we assumed the dataset was generated by a sparsemax behavior policy. As such, the in-sample constraint can be replaced by the truncation criterion. We showed two interesting facts given the assumption: (1) sparsemax policies interpolate hardmax and softmax, and when action values are fixed, consecutive sparsemax policies are within or equal to the support of the last sparsemax policy. (2) the KL divergence between a two sparsemax policies has a $q$-conditional upper bound.

We proposed Tsallis Advantage-Weighted Actor-Critic (Tsallis AWAC) that generalizes AWAC to the domain of $q$. AWAC is an important offline RL algorithm: its actor loss has been used in many state-of-the-art methods and can be derived from the perspective of KL regularization. This generalization is non-trivial: unlike AWAC that only considers maximizing log-likelihood of actions in the dataset, Tsallis AWAC introduced sparsity: it can set the loss for bad actions to zero thanks to property of the $q$-exponential formulation. Sparsity has been very recently investigated to be crucial for superior performance. Leveraging the properties of $q$-statistics, we established a $q$-conditional upper bound on the KL divergence between learned policies and the behavior policy. We also discussed another possibility of generalization by Tsallis entropy, leading to another new algorithm Tsallis InAC. We evaluated Tsallis AWAC and Tsallis InAC on the standard D4RL benchmark problem. We found that Tsallis InAC was sensitive to expert datasets on which it was among the best performer but quickly degraded on non-expert datasets. By contrast, Tsallis AWAC was more stable and outperformed AWAC by a large margin on almost every dataset, thanks to the sparsity introduced by the $q$-exponential policy. In fact, Tsallis AWAC was among the best performers in the compared algorithms.

Several interesting future directions concerning Tsallis regularization for offline RL exist. Theoretically, a probabilistic upper bound for the KL divergence of sparsemax policies and guarantees of policy improvement should be derived to deeply investigate the benefit of sparsity. Practically, it is important to improve Tsallis InAC on non-expert datasets. This could be potentially achieved by referring to Tsallis statistics literature to transform $q$-exponential. Moreover, examining the pattern of environment/dataset-specific optimal entropic index is another interesting topic.

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

## Appendix

The Appendix is organized into the following sections. We first discuss related work that helps position this paper, followed by the missing proof of Theorem 1. Lastly, implementation details and further results on Tsallis InAC are included.

## A  Related Work

### A.1  Entropy Regularization

It is a classic result that there exists a deterministic, stationary optimal policy that maximizes the cumulative return with a fixed point $Q_*(s, a)$ satisfying the Bellman optimality equation $Q_*(s, a) = r(s, a) + \gamma \mathbb{E}_{s' \sim P(\cdot|s,a)}[\max_{a'} Q_*(s', a')]$ (Puterman, 1994). The optimal policy $\pi_*$ can then be extracted by simply acting greedily with respect to the optimal action value function. In this case, $\pi_* = \arg \max_\pi \mathbb{E}_\pi [Q_*(s, a)], \forall s$ is a deterministic policy.

In this paper we consider stochastic, entropy-regularized policies as a solution of the problem $\mathbb{E}_\pi [Q(s, a) - \Omega(\pi(\cdot|s))]$, where $\Omega(\pi(\cdot|s)) \in \mathbb{R}^{|\mathcal{S}|}$ is a proper, lower semi-continuous, strictly convex function in $\pi$. Popular choices for $\Omega$ include the negative Shannon entropy $-\tau \mathcal{H}(\pi(\cdot|s)) := \tau \sum_a \pi(a|s) \ln \pi(a|s)$, which encourages the policy to be uniform with $\tau$ weighting the effect. The maximum of the problem $\max_\pi \sum_a \pi(a|s)Q(s, a) + \tau \mathcal{H}(\pi(\cdot|s))$ is attained at the well-known log-partition function $\tau \ln \sum_a \exp(\tau^{-1}Q(s, a))$ when the policy is the Boltzmann softmax distribution $\pi(a|s) \propto \exp(\tau^{-1}Q(s, a))$, where $\propto$ denotes *proportional to* up to a constant not depending on actions. KL divergence $D_{KL}(\pi(\cdot|s) \| \mu(\cdot|s)) := \sum_a \pi(a|s) \ln \frac{\pi(a|s)}{\mu(a|s)}$ is another popular choice where $\mu$ is some reference policy (Azar et al., 2012; Vieillard et al., 2020; Chan et al., 2022; Zhu & Matsubara, 2023). Using KL divergence as $\Omega$ penalizes large deviation from $\mu$. By choosing $\mu$ to be the uniform distribution, KL divergence recovers the negative Shannon entropy case. The KL-regularized optimal policy takes the form $\pi(a|s) \propto \mu(a|s) \exp(\tau^{-1}Q(s, a))$, where we overloaded the coefficient $\tau$ for Shannon entropy.

### A.2  More on Tsallis Regularization

Sparsemax policies can be obtained as a result of Tsallis regularization. Recall the sparsemax policies are written as $\pi(a|s) = \exp_q \left( \frac{Q(s,a)}{\tau} - \psi \left( \frac{Q(s,\cdot)}{\tau} \right) \right)$, $\psi \left( \frac{Q(s,\cdot)}{\tau} \right) \doteq \frac{\sum_{a \in K(s)} \frac{Q(s,a)}{\tau} - 1}{|K(s)|} + \frac{q-2}{q-1}$. Intuitively, the policy first sorts actions $a_{(1)}, \ldots, a_{(|\mathcal{A}|)}$ and then compute the threshold $\psi$. Suppose $Q(s, a_{(j+1)}) < \psi < Q(s, a_{(j)})$, then $a_{(j+1)}, \ldots, a_{(|\mathcal{A}|)}$ are truncated and have zero probability of being selected. The actions $a_{(1)}, \ldots, a_{(j)}$ are called allowable actions and collected in the set $K(s)$. The degree of truncation can be controlled by either $q$ or $\tau$. As $q$ gets larger, by definition of $\exp_q$ the truncation becomes stronger, as all $x < -\frac{1}{q-1}$ will be truncated; when $\tau$ becomes larger, more actions are collected in $K(s)$.

A nuance arises since the literature has considered different notations. We provide a comparison below in Table 2 to clarify the notations and standard used in this paper. It can be seen that this paper follows the convention of prior RL papers (Lee et al., 2020) but adopts the Taylor's first-order approximate policy from (Zhu et al., 2023). The pioneering work (Lee et al., 2018; Chow et al., 2018) can be recovered from our formulation by setting $q = 2$. Suyari & Tsukada (2005); Lee et al. (2020) did not give a computable normalization function.

| Literature | Policy Functional | General Normalization | Physics/RL |
|---|---|---|---|
| Lee et al. (2018); Chow et al. (2018) | $\left[Q(s,a)-\tilde{\psi}(Q(s,\cdot))\right]_+$ | $\tilde{\psi}:=\frac{\sum_{a\in K(s)}\frac{Q(s,a)}{\tau}-1}{|K(s)|}$ | RL |
| Lee et al. (2020) | $\left[1+(q-1)\left(Q(s,a)-\psi(Q(s,\cdot))\right)\right]_+^{\frac{1}{q-1}}$ | N/A | RL |
| Suyari & Tsukada (2005); Tsallis (2009) | $\left[1+(1-q)\left(Q(s,a)-\psi(Q(s,\cdot))\right)\right]_+^{\frac{1}{1-q}}$ | N/A | Physics |
| Zhu et al. (2023) | $\left[1+(1-q)\left(Q(s,a)-\hat{\psi}(Q(s,\cdot))\right)\right]_+^{\frac{1}{1-q}}$ | $\hat{\psi}:=\frac{\sum_{a\in K(s)}\frac{Q(s,a)}{\tau}-1}{|K(s)|}+\frac{q-2}{q-1}$ | RL |
| This paper | $\left[1+(q-1)\left(Q(s,a)-\hat{\psi}(Q(s,\cdot))\right)\right]_+^{\frac{1}{q-1}}$ | $\hat{\psi}$ | RL |

Table 2: Comparison of existing work. Physics/RL indicates the category of the cited papers. Our work follows the Taylor's first-order approximate policy of (Zhu et al., 2023), but with a change of variable $2-q$ to more consistently follow the RL literature.

### A.3 Other Regularizers and Sparse Policies

Recently, there has seen a growing body of BC/imitation-based methods featuring a variety of divergences. Ke et al. (2019) discussed imitation learning via the lens of $f$-divergence minimization and proposed variational solutions. Xu et al. (2023) proposed a framework based on the $\alpha$-divergence (Belousov & Peters, 2019) and showed that standard KL can be recovered as a special case. In the same context, they showed that Conservative Q-learning (Kumar et al., 2020) corresponds to the $\chi^2$ regularization. In this paper, we consider regularization by Tsallis KL divergence and Tsallis entropy, inducing a general class of sparsemax policies. Offline RL with sparsity has also been discussed very recently: Xu et al. (2022; 2023) showed that $\alpha$-divergence with $\alpha=-1$ also leads to sparsity. By contrast, we consider general sparsemax policies induced by Tsallis regularizers for all $q>1$. Li et al. (2023) proposed to use $q$-Gaussian distribution (Suyari & Tsukada, 2005) which is a special case of the $q$-exponential policy. In this paper we do not specify the functional form of policies, but investigating the benefits of specific $q$-exponential policy parametrizations is an interesting future direction.

## B Proof of Theorem 1

To prove Theorem 1, we decompose KL divergence as:

$$D_{KL}(\pi_t(\cdot|s)\,||\,\pi_{\mathcal{D}}(\cdot|s))=\mathbb{E}_{a\sim\pi_t(\cdot|s)}\left[\ln\pi_t(a|s)-\ln\pi_{\mathcal{D}}(a|s)\right]$$

$$=\mathbb{E}_{a\sim\pi_t(\cdot|s)}\left[\underbrace{\ln\pi_t(a|s)-\ln_q\pi_t(a|s)}_{(i)}+\underbrace{\ln_q\pi_t(a|s)-\ln_q\pi_{\mathcal{D}}(a|s)}_{(ii)}+\underbrace{\ln_q\pi_{\mathcal{D}}(a|s)-\ln\pi_{\mathcal{D}}(a|s)}_{(iii)}\right]. \tag{9}$$

Therefore, bounding respectively the three terms allows us to provide a $q$-conditional bound on the KL divergence. To bound these $\log/q$-log differences, we prove the following fact:

$$\ln x-\ln_q x=(q-1)\left[\frac{d}{dq}\ln_q x-\ln x\ln_q x\right].$$

We can verify this is true by working on the right-hand side:

$$(q-1)\left[\frac{d}{dq}\ln_q x-\ln x\ln_q x\right]=(q-1)\left[\frac{d}{dq}\frac{x^{q-1}-1}{q-1}-\ln x\ln_q x\right]$$

$$=(q-1)\left[\frac{(x^{q-1}-1)'(q-1)-(x^{q-1}-1)(q-1)'}{(q-1)^2}-\ln x\ln_q x\right]$$

$$=(q-1)\left[\frac{(q-1)x^{q-1}\ln x-(x^{q-1}-1)}{(q-1)^2}-\ln x\ln_q x\right]$$

$$=x^{q-1}\ln x-\ln_q x-(q-1)\ln x\ln_q x=((q-1)\ln_q x+1)\ln x-\ln_q x-(q-1)\ln x\ln_q x$$

$$=\ln x-\ln_q x.$$

Now to bound the log/$q$-log differences, we assume that both the learned policy and the behavior policy are sparsemax:

$$
\begin{aligned}
(i): \ln \pi_t(a|s) - \ln_q \pi_t(a|s) &= (q-1)\left[\frac{d}{dq}\ln_q \pi_t(a|s) - \ln_q \pi_t(a|s)\ln \pi_t(a|s)\right] \\
&= \pi_t^{q-1}(a|s)\ln \pi_t(a|s) - \ln_q \pi_t(a|s)\left(1 + (q-1)\ln \pi_t(a|s)\right) \\
&\leq \pi_t^{q-1}(a|s)\ln \pi_t(a|s) + \frac{1}{q-1} + \ln \pi_t(a|s) \\
&\leq \left(\pi_t^q(a|s) - \pi_t^{q-1}(a|s)\right) + \pi_t(a|s) - \frac{q-2}{q-1},
\end{aligned}
\tag{10}
$$

where we leveraged the monotonicity of $q$-logarithm; and $\ln x \leq x - 1$, $\ln_q \exp_q(x) = x$ when $x > 0$. Considering the policy is a $q$-exponential policy $\pi_t(a|s) = \exp_q\left(\frac{Q_{t-1}(s,a)}{\tau} - \psi\left(\frac{Q_{t-1}(s,\cdot)}{\tau}\right)\right)$ and $\exp_q(x) = [1 + (q-1)x]_+^{\frac{1}{q-1}}$, we must have $\pi_t(a|s) > 0 \Leftrightarrow a \in K_{t-1,q}(s) \Leftrightarrow -\frac{1}{q-1} \leq \frac{Q_{t-1}(s,a)}{\tau} - \psi\left(\frac{Q_{t-1}(s,\cdot)}{\tau}\right) \leq 0$ (Ding & Vishwanathan, 2010). **If $a \notin K_{t-1,q}(s)$, then $\ln \pi_t(a|s) = -\infty$ and the KL term is unbounded.** The same fact is exploited to yield an upper bound $\frac{1}{q-1}$ for $(ii)$. Note that the same holds true for the Tsallis KL policy $\pi_t(a|s) = \mu(a|s)\exp_q\left(\frac{Q_{t-1}(s,a)}{\tau} - \psi\left(\frac{Q_{t-1}(s,\cdot)}{\tau}\right)\right)$. Since $\ln_q x$ is monotonically increasing, we have that $-\frac{1}{q-1} \leq \ln_q \mu(a|s)\exp_q\left(\frac{Q_{t-1}(s,a)}{\tau} - \psi\left(\frac{Q_{t-1}(s,\cdot)}{\tau}\right)\right) \leq \ln_q \exp_q\left(\frac{Q_{t-1}(s,a)}{\tau} - \psi\left(\frac{Q_{t-1}(s,\cdot)}{\tau}\right)\right)$. Same by monotonicity, we can bound $(iii)$ depending on $q$ by:

$$
\begin{aligned}
(iii): \ln_q \pi_{\mathcal{D}}(a|s) - \ln \pi_{\mathcal{D}}(a|s) &= -(q-1)\left[\frac{d}{dq}\ln_q \pi_{\mathcal{D}}(a|s) - \ln_q \pi_{\mathcal{D}}(a|s)\ln \pi_{\mathcal{D}}(a|s)\right] \\
&\leq -\pi_{\mathcal{D}}^{q-1}(a|s)\ln \pi_{\mathcal{D}}(a|s) \leq -\pi_{\mathcal{D}}^{q-1}(a|s)\left(1 - \frac{1}{\pi_{\mathcal{D}}(a|s)}\right) = \pi_{\mathcal{D}}^{q-2}(a|s) - \pi_{\mathcal{D}}^{q-1}(a|s).
\end{aligned}
\tag{11}
$$

Putting all terms together, we arrive at the following $q$-dependent upper bound:

$$
D_{KL}(\pi_t(\cdot|s)\,||\,\pi_{\mathcal{D}}(\cdot|s)) \leq \begin{cases} |K_{t,q}(s)|\,\Pi_{\mathcal{D},t,q}, & \text{if } \pi_t \preceq \pi_{\mathcal{D}},\ 1 < q < \infty \\ \infty, & \text{otherwise} \end{cases}
\tag{12}
$$

where $\Pi_{\mathcal{D},t,q} := \pi_t^q(a|s) - \pi_t^{q-1}(a|s) + \pi_t(a|s) - \frac{q-3}{q-1} + \pi_{\mathcal{D}}^{q-2}(a|s) - \pi_{\mathcal{D}}^{q-1}(a|s)$, and the leading term $|K_{t,q}(s)|$ came from upper bounding the expectation $\mathbb{E}_{a \sim \pi_t(\cdot|s)}$ with all allowable actions in the set $K_{t,q}(s)$.

## C Implementation Details

We fine-tune Tsallis InAC, Tsallis AWAC and InAC since they share a same set of hyperparameters. All the algorithms used a shared set of hyperparameters found in Table 3.

A grid search was done for Tsallis InAC, Tsallis AWAC, InAC according to the same protocol as (Xiao et al., 2023). In addition, we also added a larger learning rate (0.001), which seemed to improve InAC, Tsallis AWAC, and Tsallis InAC slightly on some domains. Other methods use performance data shared by (Xiao et al., 2023). The best hyperparameters are reported in Table 4. In the grid search, we used the final 50% of evaluation tests of the normalized return to select the best hyperparameter shown. All hyperparameter settings were evaluated across 5 independent runs.

The agents train from the specified datasets, and are evaluated every 10k steps on the corresponding Mujoco environments. Normalized scores are calculated according to the min and max values provided as a part of the benchmarks (Fu et al., 2020). Finally, results are over 5 runs unless otherwise specified.

## D Details and Further Results of Tsallis InAC

**Tsallis InAC Loss Functions.** We follow in-sample softmax (Xiao et al., 2023) but replaces the Shannon entropy to Tsallis entropy. In order to fulfill the in-sample constraint, a step similar to (Xiao et al., 2023) is

| Name | Value |
|---|---|
| Number of steps | 1000000 |
| Logging interval | 10000 |
| Hidden Units | 256 |
| Batch Size | 256 |
| Target Network Update Rate | 1 |
| Polyak Constant | 0.995 |
| Discount ($\gamma$) | 0.99 |
| Learning Rate | swept |
| Regularization coefficient ($\tau$) | swept |

| Name | Swetp Values |
|---|---|
| Learning Rate | [0.00003, 0.0001, 0.0003, 0.001] |
| $\tau$ | [0.01, 0.1, 0.33, 0.5, 1.0] |

Table 3: (Left) Shared hyperparameters. (Right) Swept hyperparameters.

| Environment | Dataset | Parameter | Tsallis InAC | Tsallis AWAC |
|---|---|---|---|---|
| Ant | Expert | lr | 0.01 | 0.01 |
| | | $\tau$ | 0.5 | 1.0 |
| | Medium Expert | lr | 0.0003 | 0.001 |
| | | $\tau$ | 0.1 | 1.0 |
| | Medium Replay | lr | 0.0003 | 0.001 |
| | | $\tau$ | 0.01 | 0.5 |
| HalfCheetah | Expert | lr | 0.001 | 0.001 |
| | | $\tau$ | 0.33 | 1.0 |
| | Medium Expert | lr | 0.001 | 0.001 |
| | | $\tau$ | 0.5 | 1.0 |
| | Medium Replay | lr | 0.001 | 0.001 |
| | | $\tau$ | 1.0 | 0.01 |
| Hopper | Expert | lr | 0.001 | 0.001 |
| | | $\tau$ | 0.01 | 1.0 |
| | Medium Expert | lr | 0.0001 | 0.001 |
| | | $\tau$ | 0.01 | 0.5 |
| | Medium Replay | lr | 0.001 | 0.001 |
| | | $\tau$ | 0.01 | 0.5 |
| Walker | Expert | lr | 0.001 | 0.001 |
| | | $\tau$ | 0.1 | 1.0 |
| | Medium Expert | lr | 0.0003 | 0.001 |
| | | $\tau$ | 0.01 | 0.01 |
| | Medium Replay | lr | 0.001 | 0.001 |
| | | $\tau$ | 0.01 | 0.5 |

Table 4: Best learning rate (lr) and regularization coefficient $\tau$ for the proposed method.

made to extract $\pi_{\mathcal{D}}$ from the Tsallis entropy regularized policy for the actor loss:

$$
\begin{aligned}
\hat{\pi}_{Q_\theta,\pi_{\mathcal{D}}}^{\texttt{TInAC}}(a|s) &\propto \mathbb{1}\{a : \pi_{\mathcal{D}}(a|s) > 0\} \cdot \exp_q\left(\frac{1}{\tau}Q_\theta(s,a)\right) \\
&= \pi_{\mathcal{D}}(a|s)\pi_{\mathcal{D}}(a|s)^{-1}\exp_q\left(\frac{1}{\tau}Q_\theta(s,a)\right) = \pi_{\mathcal{D}}(a|s)\exp_q\left(\ln_q\frac{1}{\pi_{\mathcal{D}}(a|s)}\right)\exp_q\left(\frac{1}{\tau}Q_\theta(s,a)\right) \\
&= \pi_{\mathcal{D}}(a|s)\left(\exp_q\left(\frac{1}{\tau}Q_\theta(s,a) + \ln_q\frac{1}{\pi_{\mathcal{D}}(a|s)}\right)^{q-1} + (q-1)^2\frac{1}{\tau}Q_\theta(s,a)\ln_q\frac{1}{\pi_{\mathcal{D}}(a|s)}\right)^{\frac{1}{q-1}}.
\end{aligned}
\tag{13}
$$

where we assumed that $\frac{0}{0} = 0$. This can happen when $Q_\theta(s,a) = 0$ and $\pi_{\mathcal{D}} = 0$. In this case, the entire Eq. (13) should be 0 due to conditional on $\pi_{\mathcal{D}}$. $\left(\exp_q x \cdot \exp_q y\right)^{q-1} = \exp_q(x+y)^{q-1} + (q-1)^2 xy$ (Yamano, 2002). Following (Haarnoja et al., 2018), we update towards this policy by minimizing the KL divergence

between $\hat{\pi}_{Q_\theta,\pi_\mathcal{D}}^{\texttt{TInAC}}$ and $\pi_\phi$. When used as the first argument of KL loss, the leading $\pi_\mathcal{D}$ allows us to compute the loss using only actions from the dataset:

$$D_{KL}\left(\hat{\pi}_{Q_\theta,\pi_\mathcal{D}}^{\texttt{TInAC}}(\cdot|s)\,||\,\pi_\phi(\cdot|s)\right) = \mathbb{E}_{a\sim\hat{\pi}_{Q_\theta,\pi_\mathcal{D}}^{\texttt{TInAC}}(\cdot|s)}\left[\ln\hat{\pi}_{Q_\theta,\pi_\mathcal{D}}^{\texttt{TInAC}}(a|s) - \ln\pi_\phi(a|s)\right]$$

$$= \mathbb{E}_{a\sim\pi_\mathcal{D}(\cdot|s)}\left[\left(\exp_q\left(\frac{Q_\theta(s,a)}{\tau} + \ln_q\frac{1}{\pi_\omega(a|s)}\right)^{q-1} + (q-1)^2\frac{Q_\theta(s,a)\ln_q\frac{1}{\pi_\omega(a|s)}}{\tau}\right)^{\frac{1}{q-1}}\left(\ln\hat{\pi}_{Q_\theta,\pi_\mathcal{D}}^{\texttt{TInAC}}(a|s) - \ln\pi_\phi(a|s)\right)\right].$$

Notice the term $\ln\hat{\pi}_{Q_\theta,\pi_\mathcal{D}}^{\texttt{TInAC}}(a|s)$ does not depend on $\pi_\phi$ and hence can be removed from the actor loss. We observed that for Tsallis entropy regularization, adding normalization tends to underperform, therefore we propose to remove the normalization. We write out our losses for $q = 2$ :

$$\mathcal{L}_{\text{actor}}^{\texttt{TInAC}}(\phi) = -\mathbb{E}_{s,a\sim\mathcal{D}}\left[\left(\left(\exp_2\left(\frac{Q_\theta(s,a)}{\tau} + \ln_2\frac{1}{\pi_\omega(a|s)}\right) + \frac{Q_\theta(s,a)\ln_2\frac{1}{\pi_\omega(a|s)}}{\tau}\right)\ln\pi_\phi(a|s)\right],$$

$$\mathcal{L}_{\text{critic}}^{\texttt{TInAC}}(\theta) = \mathbb{E}_{s,a,r,s'\sim\mathcal{D}}\left[(r + \gamma V_\zeta(s') - Q_\theta(s,a))^2\right], \tag{14}$$

$$\mathcal{L}_{\text{baseline}}^{\texttt{TInAC}}(\zeta) = \mathbb{E}_{s\sim\mathcal{D},a\sim\pi_\phi(\cdot|s)}\left[(V_\zeta(s) - (Q_\theta(s,a) - \tau\ln_2\pi_\phi(a|s)))^2\right].$$

The term $\frac{1}{\pi_\omega(a|s)}$ in $\mathcal{L}_{\text{actor}}^{\texttt{TInAC}}(\phi)$ is likely to cause numerical issues. To avoid it, we clip the range of $\pi_\omega$ by $\max\{\epsilon,\pi_\omega(a|s)\}$, with $\epsilon = 10^{-8}$. However, since $\ln_q x$ is proportional to the $q$-th power, we are unable to sweep over larger $q$ due to numerical problems even with clipping. Therefore, in the experiments we sweep different $q$ for the Tsallis KL implementation only.

**Remark.** Let us focus on $\mathcal{L}_{\text{actor}}^{\texttt{TInAC}}(\phi)$ and define the term in the bracket before $\ln\pi_\phi$ as $C$:

$$C := \exp_2\left(\frac{Q_\theta(s,a)}{\tau} + \ln_2\frac{1}{\pi_\omega(a|s)}\right) + \frac{Q_\theta(s,a)\ln_2\frac{1}{\pi_\omega(a|s)}}{\tau},$$

then $\mathcal{L}_{\text{actor}}^{\texttt{TInAC}}(\phi) = -\mathbb{E}_{s,a\sim\mathcal{D}}\left[C\ln\pi_\phi(a|s)\right]$. Notice that $\exp_2\left(\frac{Q_\theta(s,a)}{\tau} + \ln_2\frac{1}{\pi_\omega(a|s)}\right)$ can be written as

$$\mathbb{1}\left\{1 + \frac{Q_\theta(s,a)}{\tau} + \ln_2\frac{1}{\pi_\omega(a|s)} > 0\right\}\cdot\left(1 + \frac{Q_\theta(s,a)}{\tau} + \ln_2\frac{1}{\pi_\omega(a|s)}\right),$$

therefore, it must be non-negative and the sign of $C$ depends on $\frac{Q_\theta(s,a)\ln_2\frac{1}{\pi_\omega(a|s)}}{\tau}$. However, since $\ln_2\frac{1}{\pi_\omega(a|s)} > 0$, the sign actually depends on $Q_\theta(s,a)$ only. If this term is negative and $\left|\frac{Q_\theta(s,a)\ln_2\frac{1}{\pi_\omega(a|s)}}{\tau}\right| > \exp_2\left(\frac{Q_\theta(s,a)}{\tau} + \ln_2\frac{1}{\pi_\omega(a|s)}\right)$, then we will be minimizing the loss $\mathbb{E}_{s,a\sim\mathcal{D}}\left[|C|\ln\pi_\phi(a|s)\right]$, which corresponds to thinking an action $a$ is bad (since $Q_\theta(s,a)$ is highly negative) and explicitly minimizing its log-likelihood; otherwise, we will be maximizing its likelihood. By contrast, all exponential-based methods (exponential advantage weighted regression algorithms like AWAC) do not minimize likelihood for actions in the dataset explicitly, since exp is always positive. Moreover, this suggests that the behavior of Tsallis InAC may be more extreme with larger $q$, since by equation 13, $(q-1)^2\frac{Q_\theta(s,a)\ln_2\frac{1}{\pi_\omega(a|s)}}{\tau}$ will have a large negative value when $q$ is large even with slightly negative $Q_\theta(s,a)$. As a summary, Tsallis InAC may behave drastically: it may perform either very well or very bad depending on whether the dataset contains a majority of near-expert trajectories.

**Tsallis InAC against InAC.** From Figure 8 it is clear that Tsallis InAC competes favorably against InAC only on the Expert dataset, and degrades significantly with the increase of non-expert trajectories in the dataset. In fact, Tsallis InAC behaves like a behavior cloning method and learns very fast on Expert `HalfCheetah` and `Hopper`. As analyzed in the remark after equation 14, the difference between InAC and AWAC policies lies in the term $-\ln\pi_\mathcal{D}$ in the policy $\pi^{\texttt{InAC}}(a|s) \propto \pi_\mathcal{D}(a|s)\exp\left(\frac{Q_\theta(s,a)}{\tau} - \ln\pi_\mathcal{D}(a|s)\right)$. This term can correct the bias introduced by suboptimal data distributions: a poor action with high $\pi_\mathcal{D}(a|s)$

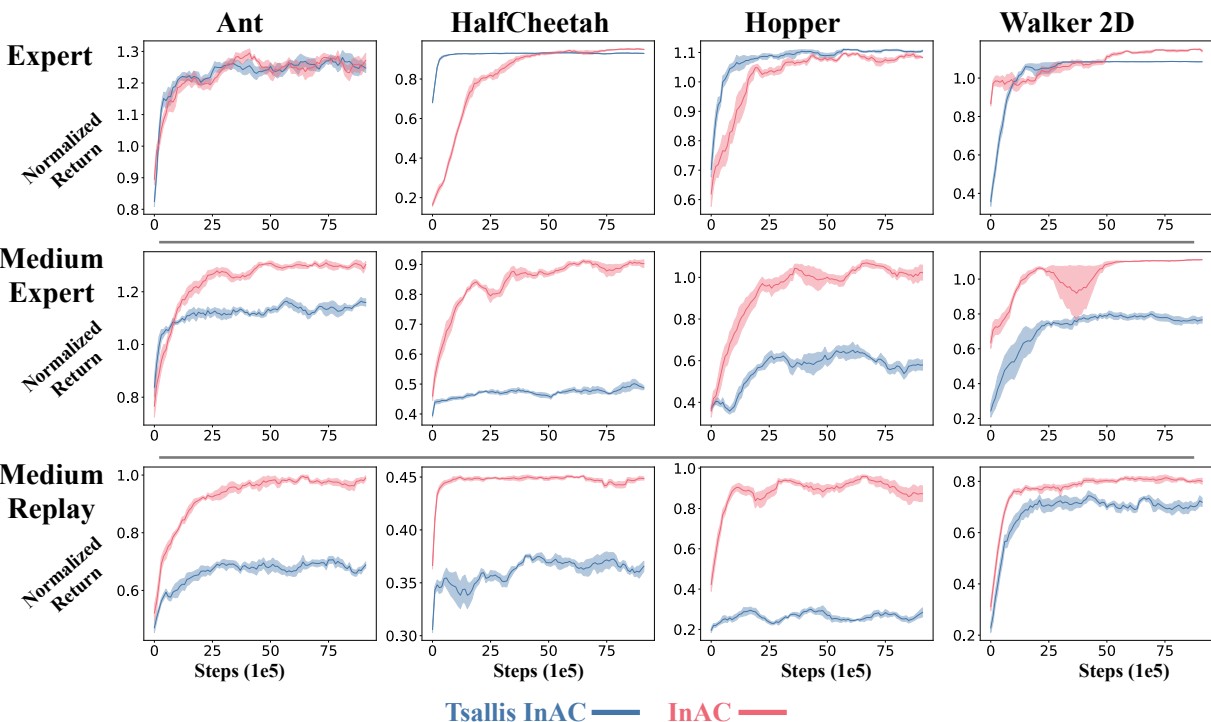

Figure 8: Comparison between Tsallis InAC and InAC. Normalized policy evaluation scores on Mujoco Expert and Medium Expert datasets over one million steps. Results are the average over 5 runs with ribbon denoting the standard error. $y$-axis: scores; $x$-axis: number of iterations.

does not necessarily lead to high $\pi^{\texttt{InAC}}(a|s)$ since it is weighted down inside the exponential by $-\ln \pi_{\mathcal{D}}(a|s)$. While Tsallis InAC follows this design choice, in general $\ln_q \frac{1}{\pi_{\mathcal{D}}(a|s)} \gg \ln \frac{1}{\pi_{\mathcal{D}}(a|s)}$. In fact, since $\ln_q \frac{1}{\pi_{\mathcal{D}}(a|s)}$ is proportional to the $q$-th power of $\frac{1}{\pi_{\mathcal{D}}(a|s)} \geq 1$, this term is likely to dominate the entire $\mathcal{L}^{\texttt{TInAC}}_{\text{actor}}$, suggesting Tsallis InAC is very sensitive to the level of behavior policy like a BC method.

