# OpenReview forum: "Offline Reinforcement Learning via Tsallis Regularization"
_TMLR — Accepted by TMLR_

### Review · Reviewer_qDjc · 2024-02-23

**Summary Of Contributions:**

There are numerous methods being developed to learn policies with offline static datasets. Among them, one of the most popular methods is policy regularization method, which actively constrain the learned policy to be close to the behavior policy (aka, data-collecting policy). This paper lies in this category. The authors propose Tsallis regularization for offline reinforcement learning (RL), which realizes in-sample policy constraint via sparsemax method. The authors show that the learned sparsemax policies may have sparsity-conditional KL divergence to the behavior policy. Moreover, the authors combine their Tsallis regularization with two methods, InAC and AWAC and empirically evaluate their performance in the D4RL datasets.

**Audience:**

Yes

**Broader Impact Concerns:**

I believe no broader impact concerns are needed to be specified.

**Claims And Evidence:**

Yes

**Requested Changes:**

Please answer my above questions and modify the manuscript accorrdingly. I also have the following additional comments:

- reorganize section 4 and summarize a formal lemma/proposition/theorem. The proof details can be attached in the appendix.

- the following work seems to be relevant (in-sample offline learning) and should be included in the manuscript [1]

[1] State Advantage Weighting for Offline RL

**Strengths And Weaknesses:**

=== Strengths ===

- The paper is mostly well-written, and I appreciate how the authors carefully explain their motivation, supporting it with empirical evidence.

- The resulting method is simple. It is also nice to see that the authors combine their method with a recent work, InAC.

- detailed parameter study is conducted

=== Weaknesses ===

- the novelty of this paper is somewhat limited, but this is not a big issue for this venue

- some expressions are confusing, I list some of them below:

  - page 1, *Since the environment cannot be sampled, the bias can never be alleviated by visiting the region and correcting the value estimate*, what do you mean by the environment cannot be sampled? I also disagree that the value estimate cannot be corrected with only static data. This can be done with uncertainty based methods like UWAC [1]

  - page 2, *Intuitively, this assumption invites learning an improved sparsemax policy with support within the that of the behavior policy*, this sentence is unnecessarily complex. Also, what do you mean by *within the that of the behavior policy*? Is this sentence generated by LLMs? If this is generated by LLMs, please make sure that all of the sentences are checked by humans. Meanwhile, please break down long, complex sentences into short, concise sentences in the entire paper.

[1] Uncertainty weighted actor-critic for offline reinforcement learning

- the quality of the figures can be significantly improved (e.g., Figure 1). Please try to export pdf with matplotlib instead of taking screenshots by convention.

- Tsallis AWAC beats AWAC on most of the tasks, however Tsallis InAC underperforms InAC on most of the tasks. The authors comment that this is due to *Tsallis InAC is very sensitive to the level of behavior policy like a BC method* which is analyzed in the remark after equation 12. I do not seem to find the corresponding explanations and analysis. Please make it clearer. Also, please add discussions on the limitations of your method.

- it is vital to analyze the applicability of your method. Since it downgrades the performance of InAC in numerous tasks, is it general enough to benefit a wide range of policy regularization offline RL algorithms?

- no code seems to be provided

- the performance reported in this paper is obtained by a very careful tuning of hyperparameters (see Table 3) . This further downgrades the applicability of the proposed method. Meanwhile, please specify detailed hyperparameter setup on each dataset.

- learning rate seems to be a critical hyperparameter, but no parameter study is conducted

- the experiments are only conducted on D4RL MuJoCo datasets. It is unclear how the proposed method behave on another domain like antmaze, adroit, etc. It is also unclear whether the proposed method can achieve good performance on medium-level datasets. More experiments are expected, which I believe will make this paper stronger. I know this means a large number of experiments, please at least conduct experiments on 2 antmaze datasets, 2 adroit datasets, and medium-level datasets

- the compared baselines are few, please consider adding more baselines in Figure 6. If you do not have time to run them, you can create a table and directly take the reported numerical results of recent baselines from their papers.

---

> ### Author Response · Authors · 2024-03-25
> **Thank you for the comments**
>
> **The novelty of this paper is somewhat limited, but this is not a big issue for this venue**\
> We believe the novelty of this paper lies in establishing a novel link between the Tsallis truncation to the offline RL paradigm, which is radically different from online Tsallis algorithms. As a result, more general loss functions featuring sparsity induced by q-exponentials, and a bound on policy difference are derived.
>
> **Some expressions are confusing.**
> 1. We corrected our phrasing by “since no further interactions with the environment is allowed, the agent may be increasingly prone to choose actions with delusively high values”.
> 2. We apologize for the confusing long sentence here. However, we would like to point out the manuscript was completely written by the authors rather than LLMs. The sentence is rephrased as “Intuitively, the assumption suggests to learn a policy that shrinks the support of the behavior policy”
>
> **The quality of the figures can be significantly improved (e.g., Figure 1). Please try to export pdf with matplotlib instead of taking screenshots by convention.**\
> We have replaced png files to PDF.
>
> **Tsallis AWAC beats AWAC on most of the tasks, however Tsallis InAC underperforms InAC on most of the tasks. I do not seem to find the corresponding explanations and analysis. Please make it clearer.**\
> We apologize for the vagueness. Tsallis InAC is now only briefly introduced and its derivation, analysis and performance have all been moved to Appendix D. In Section 5 we have modified the introductory text for Tsallis InAC.
>
> **Please add discussions on the limitations of your method.**\
> We have added new *Section 7 Limitation of The Proposed Method* for discussing the limitations.
>
> **It is vital to analyze the applicability of your method. Since it downgrades the performance of InAC in numerous tasks, is it general enough to benefit a wide range of policy regularization offline RL algorithms?**\
> As pointed out by both reviewers, Tsallis InAC seems to be applicable to expert level datasets. We have decided to move it to the appendix, and emphasize more the contribution of Tsallis AWAC. In the new Table 1 it can be seen that Tsallis AWAC shows positive overall performance.
>
> **No code seems to be provided**\
> We apologize for having missed the source code, it is now attached in the revised version.
>
> **The performance reported in this paper is obtained by a very careful tuning of hyperparameters (see Table 3) . This further downgrades the applicability of the proposed method. Meanwhile, please specify detailed hyperparameter setup on each dataset.**\
> We would like to point out that all the compared algorithms were evaluated using care tuning of hyperparameters as per their publications. Applicability without fine-tuned parameters is beyond the scope of this paper and is left as an interesting future research direction.
>
> Regarding individual hyperparameters, we have included a new table Table 4 for this purpose.
>
> **Learning rate seems to be a critical hyperparameter, but no parameter study is conducted**\
> We have added a new table Table 4 for the best learning rate and entropy coefficient swept for individual dataset, obtained from a grid search. Another parameter study was shown in Figure 5 where the interplay between parameters $q$ and $\tau$ was studied.
>
> **the experiments are only conducted on D4RL MuJoCo datasets. It is unclear how the proposed method behave on another domain like antmaze, adroit, etc. Please at least conduct experiments on 2 antmaze datasets, 2 adroit datasets, and medium-level datasets**\
> We thank the reviewer for pointing out potential new environments. However, we believe the current results are sufficient to support the claim of the paper:
> We are the first to introduce Tsallis regularization to offline RL and analyze them both theoretically and empirically. Experimental results are in support of our theoretical findings:
> 1. - Tsallis AWAC improved upon AWAC due to the $q$-exp function and Tsallis KL regularization. In fact Tsallis AWAC was among the best performers among the compared baselines
>    - ablation study on $q/\tau$ manifested that sparsity indeed plays a crucial role in the performance
>    -  KL divergence plot shows convergence for different $q$ which is consistent with Section 4.
> 2. We did not claim Tsallis regularization to be generally applicable. In fact, we emphasized that “this paper is positioned in the BC/imitation learning methods literature”. Therefore, we believe the existing results are supportive of this claim.  This paper serves as the beginning and foundation of future investigation for more performant and generally applicable methods.
>
> **the compared baselines are few, please consider adding more baselines in Figure 6.**\
> We have included the new Table 1 to compare against new baselines taken from the suggested paper “State Advantage Weighting for Offline RL”.

---

> > ### Comment · Reviewer_qDjc · 2024-03-26
> >
> > Thanks for answering my questions. The current version looks better than the previous one. Some minor points:
> >
> > - it is a pity that no new results on antmaze or adroit datasets are presented, but I can understand that since it requires a lot of time. It is still my hope that the authors could add at least some results on them in the final version because MuJoCo datasets are simple and the performance on them is saturated. I am also okay if the authors do not do so mainly due to the fact that the authors have already conducted numerous experiments, which I believe is sufficient for publication
> >
> > - next time, it is better to highlight the revised contents in color (e.g., blue /red) to aid the reviewers better capture the changes in the manuscript.
> >
> > All in all, a nice work!

---

> ### Author Response · Authors · 2024-03-26
>
> We thank your again for the helpful feedback. We will take these valuable suggestions into account for our next submission.

---

### Review · Reviewer_pk3h · 2024-03-16

**Summary Of Contributions:**

This paper presents Tsallis-regularized algorithms for offline RL. The main idea is to translate some recent ideas from Zhu et al. (2023) in the online setting that takes standard RL algorithms that use KL divergences and replace them with Tsallis divergences. This idea is applied on top of the AWAC and InAc algorithms for offline RL with promising results on top of AWAC and less promising results on top of InAc.

**Audience:**

Yes

**Claims And Evidence:**

Yes

**Requested Changes:**

To resolve the weaknesses above, I would suggest the following changes:

1. Clarify the discrete vs. continuous issue.

2. Improve the presentation of the theoretical section and connect it to the proposed algorithms more clearly.

3. Streamline the presentation of sections 2 and 3.

4. Move the InAc-related results to an appendix and focus more on the AWAC-related results. If the authors/other reviewers think this is too strong, I would at least advocate separating the paper to present these as two more clearly distinct algorithms by e.g. first presenting the AWAC algorithms and experimental results together and then presenting the InAc algorithm and experiments second.

**Strengths And Weaknesses:**

### Strengths

1. The proposed algorithm is simple and elegant as well as novel for the offline RL setting. The idea of using Tsallis regularizers that induce sparsemax instead of softmax policies (as in AWAC) is intuitively appealing for the offline RL setting.

2. The experiments are thorough, providing a variety of sweeps on standard benchmark tasks as well as some diagnostic experiments to understand the behavior of the proposed algorithms.

### Weaknesses

1. There is a disconnect between the presentation and the actual algorithms implemented for the experiments since to my understanding, the presentation is entirely for discrete action spaces, but the algorithms are applied in continuous spaces. This major disconnect seems to be only briefly addressed in the algorithm presentation when certain terms are replaced to allow for continuous actions. Since this likely significantly affects the performance of the algorithms in the experimental results, more clarity is needed as to how the motivation connects to the actually implemented algorithms.

2. The theoretical section (section 4) is somewhat confusing. First, making a more clear theorem-proof structure could help to clarify what the main assumptions are and what the takeaway is supposed to be for the reader. Second, it is unclear to me what this section is supposed to be doing. It seems that the idea is to relate the proposed sparsemax policy to the standard KL divergence in related work, but it is not clearly introduced and the final result is very difficult to interpret.

3. The paper overall is a bit verbose. While I understand that the authors may want to provide detailed introduction, it could be useful to get into the novel contributions more quickly.

4. I am confused by the decision to present the largely negative results for Tsallis-InAc on similar footing to the positive results for Tsallis-AWAC. It is nice to present the negative results, but it could perhaps streamline the presentation and takeaways for the reader to focus the paper on the modifications to AWAC and move the InAc-related results to an appendix for the interested reader.

Minor comments:
- The sum at the end of the first paragraph at the top of page 2 is referred to as ``softmax'', but it not in fact the standard softmax function (which is actually a soft version of argmax). I know this is not a specific problem with this paper, but with terminology in the field at large, but to reduce confusion it could be worth either referring to this quantity as sumexp (or logsumexp) where appropriate, or to add some footnote just clarifying this is not the standard soft arg max.
- The axes labels in figure 1 are too small.

---

> ### Author Response · Authors · 2024-03-25
> **thank you for the comment**
>
> We thank the reviewer for providing valuable comments on structuring the paper. We followed the advice to streamline the paper into 12 pages. Details are provided in the following:
>
> **Clarify the discrete vs. continuous issue.**\
> We added the following paragraph to the background section to clarify the mismatch between theory and practice: \
> *It is worth noting that offline RL problems often have continuous action spaces, while many existing methods were derived based on a discrete footing. Approximation is usually necessary, e.g., in evaluating policies (Fujimoto et al., 2019; Xiao et al., 2023). Similar to (Xiao et al., 2023), our method is positioned in the entropy-regularized literature and established based on the discrete action setting (Vieillard et al., 2020). When applied to continuous action problems, our method also necessitates approximation in policy evaluation, as detailed in Section 5. While such approximation may affect the performance, its use is justified in the following sense: (1) evaluating the continuous policy exactly is generally intractable; (2) the approximate policy is still an exp/q-exp function which retains some desired properties. Developing theories in the continuous action setting is left as an interesting future work.*
>
> **Improve the presentation of the theoretical section and connect it to the proposed algorithms more clearly.**\
> We have made a new Theorem 1 to state the main point and moved all technical details to the Appendix. The following paragraph is added to before Theorem 1 to summarize the motivation: \
> The following theorem states that the upper bound on KL divergence between learned/behavior policies
> can be flexibly adjusted by changing q. Therefore, our prior knowledge regarding the quality of the dataset may impact our choice of q: when we are confident that sufficient near-expert trajectories exist, it may be preferable to choose a large q to encourage more similarity to $\pi_{\mathcal{D}}$ ; on the other hand, choosing q = 1 may be more robust to a non-expert dataset
>
> **Streamline the presentation of sections 2 and 3.**\
> We have largely streamlined Section 2 and 3 to highlight the contribution. Relevant materials have been moved to the Related Work section in the Appendix.
>
> **Move the InAc-related results to an appendix and focus more on the AWAC-related results.**\
> Tsallis InAC now is only briefly introduced in Section 5 to emphasize Tsallis AWAC more. Its derivation, analysis and performance have been moved to the Appendix. In Section 5 we modified Tsallis InAC as:\
> *Tsallis In-Sample Actor-Critic (Tsallis InAC). Different from Tsallis AWAC that has a clean policy
> form, Tsallis entropy regularization induces a much more complicated policy (see Appendix D)
> It can be seen that the Tsallis InAC policy cannot contain everything inside the q-exp function.
> This can lead to numerical issues and more importantly, the loss of sparsity. We found that Tsallis InAC in general underperforms Tsallis AWAC. Two potential reasons are identified. Firstly, the Tsallis InAC actor loss in general does not lead to sparsity, since there is a $Q_{\theta}(s,a) \ln_2 \frac{1}{\pi_{\omega}}(a|s)$ term outside the q-exponential function. This term is very sensitive to the sign of Q and magnifies its value by the term $\ln_2 \frac{1}{\pi}$ which is typically large. Directly from this, the second point is that Tsallis InAC may be drastic in deciding good/bad actions simply by its sign and can be minimizing likelihood of bad actions. This stands in sheer contrast to InAC and Tsallis AWAC, which do not explicitly minimize likelihood. The drastic behavior of Tsallis InAC in general leads to undesired performance, but with exception for expert datasets, see Appendix D.*

---

### Review · Reviewer_XgiD · 2024-03-20

**Summary Of Contributions:**

The paper considers modifications of two offline RL algorithms in-sample actor-critic (InAC) and advantage-weighted actor-critic (AWAC): Tsallis InAC, based on Tsallis Entropy, and Tsallis AWAC, based on Tsallis KL divergence. The paper investigates various problem statements, compares them with previous works, and provides theoretical insights. The loss functions are stated, and the algorithms are tested for four Mujoco tasks.

**Audience:**

Yes

**Claims And Evidence:**

No

**Requested Changes:**

See the sections above.

**Strengths And Weaknesses:**

Strengths:
* The paper demonstrates that a seemingly simple replacement of a regularization function can have a big impact on policy behavior.
* The paper has several interesting findings concerning sparsity, the role of regularization on various stages of the algorithm, behavior for expert and non-expert datasets, or a bound on the KL divergence between two sparsemax policies.
* The paper connects the dots between multiple approaches existing in the literature.

Weaknesses:
* General comments:
  * Sections 3-5 are dense and contain a lot of inline formulas. The content of these sections could be organized in a more clean way. Additionally, the paper would be much easier to read if it was (close to) self-contained. The major results used from other papers should be proved (or sketched) in the Appendix.
  * It seems that the same results are repeated in several places. It would be nice to have the main facts gathered in one place (e.g., the solutions to the optimization problems under various regularization terms).
  * There is no related work section; the references are scattered across the paper.
  * The paper could be shortened by moving parts of the math to the Appendix and reducing the repetitive statements (e.g., Figs 4-6, or the form of the solution to regularized problem).
  * The environments are simple Mujoco tasks and there are not that many offline RL baselines.

* More details follow:
  * Section 2:
	* In the paragraph following eq. (1), the characterization of $K(s)$ does not clearly match the one in the cited reference (Zhu et al. 2023, page 15).
	* Equation (1) describes approximation, but it is not clear to what and in what sense.
  * Section 3:
	* In the last paragraph of page 5 it is not clear where $\mathcal H$ is (does not appear in eqs. (2) or (3)).
	* In eq (4), there should either be $\pi_*$ or $Q_{t, \pi_\mathcal D}$.
	* The discussion around eq. (5) is unclear, particularly around $K_{t,2}$. Additionally, only $K$ depends on $t$.
	* The last paragraph of this section discusses using regularization for policy improvement and policy evaluation. It is unclear, whether the choice of where to use regularization follows from the method or is a design choice. In the latter case, there would be the question concerning the methods' convergence.
  * Section 4:
	* This section should be structured more formally, with the main result (eq. (9)) as a theorem with all the necessary assumptions and the rest of the section as proof. The proof could be placed in the Appendix.
  * Section 5:
	* The last paragraph of Section 3 talks about the regularization of policy improvement and policy evaluation, and hints at more details in Section 5. This topic should be expanded upon in Section 5, with a clearer explanation of where the regularization enters the Tallis InAC and Tallis AWAC loss functions.
	* Eq. (11) should assume some convention when dealing with for expressions like 0/0.
	* The term "baseline" is overloaded: there are "baseline algorithms" and a "baseline" in the advantage function (used in the subscript for the losses).
  * Section 6:
	* Environments are simple Mujoco tasks (Ant, HalfCheetah, Hopper, Walker 2D).
	* Fig 3 should be placed in the Appendix.
	* The datasets expert, medium expert, and medium replay should be explained in more detail (in particular, medium replay is not explained).
	* There is a small number of seeds (5).
	* Figures 4-6 duplicate a lot of information. The redundancy should be reduced. Additionally, making the colors on the plot more distinct could help with the figures' readability.
  * Section 6.2
	* Why only include Hopper and Walker2d (in expert versions)?
	* Some statements are not very clear:
		* "larger q tend to learn quicker and are relatively insensitive to training" - seems rather vague and not necessarily visible in Figure 7.
		* "shrinks the gap Q-V".
		* "larger set of allowable actions".
	* Consider splitting Fig 8 and Fig 9: Fig 9 belongs to Sec. 6.2, Fig 8 to Sec 6.3.

Questions:
* How does the behavior of methods for near-expert and non-expert data relate to the findings in [1]?

[1] Kumar, et al. When Should We Prefer Offline Reinforcement Learning Over Behavioral Cloning?, 2022

---

> ### Author Response · Authors · 2024-03-25
> **thank you for the detailed comments**
>
> **Section 2**:
> 1. We thank the reviewer for pointing out the typo. We have corrected it and provided a new Table 2 in the appendix to help better clarify the notations used in this paper.
> 2. We have modified the explaining text as follows:\
> *For $q\neq 1,2,\infty$, the resulting policy does not have a closed-form expression, since the normalization function is a sum of radicals. But we can apply Taylor's first-order expansion  on the resulting $q$-exp function to obtain an approximate policy form (Zhu et al., 2023)*
>
> **Section 3**:
> 1. $\mathcal{H}$ originally appeared in the introductory text in Section 2, before Section 2.1. However, we recognize that it was likely to reduce readability, therefore we have removed it from Section 3.
> 2. We have fixed the issue by choosing $Q_{t, \pi_D}$.
> 3. We have modified Q to be dependent on t as well. Regarding eq.(5) (new eq.(3)), we have modified the discussion as follows:\
> *Continuing the discussion of Eq.(3), let us define $K_{0, q}(s) := K_\mathcal{D}(s)$ as our starting point. Then during learning,  a new subset of allowable actions $K_{t, q}(s)$ depending on $q$ and iteration $t$ can be extracted from $K_{0, q}(s)$. Recursively, every $K_{t+1, q}$ contains only a subset of actions from the last set $K_{t, q}\,$, or mathematically speaking,  $K_{0, q}\succeq K_{1,q} \succeq K_{2,q} \succeq \dots \succeq K_{t,q} \succeq K_{t+1, q}$, where we used $A \succeq B$ to denote $A$ is a subset of $B$.*
> 4. We have added new Eq.(5) in policy iteration form to help clarify where regularization enters.
>
> **Section 4**:
> 1. We have restructured the section into the theorem-proof style and moved details to Appendix B.
>
> **Section 5**:
> 1. To help clarify where the regularization enters, we have added a new Eq. (5) to explain the difference between Tsallis InAC and Tsallis AWAC.
> 2. We have added the following explanation after Eq. (13):\
> *where we assumed that $\frac{0}{0} = 0$. This can happen when $Q_{\theta}(s,a) = 0$ and $\\pi_{\mathcal{D}} = 0$. In this case, the entire Eq.(13) should be 0 due to conditional on $\pi_{\mathcal{D}}$.*
> 3. We have replaced the phrasing “baseline algorithms” to “compared algorithms” or “existing algorithms”
>
> **Section 6**:
> 1. We wish to emphasize that we do not claim Tsallis regularization to be generally applicable. In fact, we emphasize that “this paper is positioned in the BC/imitation learning methods literature”. Using Tsallis regularization in more challenging environments could entail more sophisticated tricks and extensive tuning, and this paper serves as the foundation of future investigation,  which is the first to introduce Tsallis regularization to RL and establish its formulation.  In these simple, proof-of-concept environments, our results support our claims. In fact, Tsallis AWAC is strong by comparison. Ablation studies also provide evidence that the framework is worth further investigating.
> 2. We have moved all experimental details to Appendix C
> 3. We have expanded the explanation of datasets.
> 4. While increasing the number of seeds would improve the confidence of our results, we still show statistically significant results within the number of seeds used. We also follow standard practices in the field. For example, CQL uses 4 seeds only, TD3BC and BRAC use 5 seeds.
> 5. Our proposed methods Tsallis AWAC (resp. Tsallis InAC) can be seen as an extension of AWAC (resp. InAC). Therefore we believe it is necessary to separate their comparison from the overall comparison. However,  we do acknowledge that they duplicated much information. We have moved the Tsallis InAC figure to the appendix, and added a new Table 1 for new comparison. \
> Regarding the color scheme, we took color-blindness into account following https://cran.r-project.org/web/packages/khroma/vignettes/tol.html#muted. Therefore we have decided to keep the current color scheme.
>
> **Section 6.2**
> 1. By the sensitivity study we try to find evidence in support of our claim:
> “We position this paper in the popular BC/imitation-based literature, where the goal of learning is to reproduce the near-expert behavior policy”. Therefore we focus on expert data, since performance on other datasets can be seen from Table 1 and Figure 5. Results on all expert environments are similar, we show Hopper and Walker2d since they are representative and we would like to reduce redundancy.
> 2. we have modified the text accordingly.
>
>
> **Question**:
> How does the behavior of methods for near-expert and non-expert data relate to the findings in [1]?
>
> Table 1 allows us to compare our methods to BC and TD3BC. On medium datasets BC, TD3BC and Tsallis InAC perform poorly. This may reflect the findings in (Kumar et al., 2022), especially their Practical observations 4.1, 4.2. But it is worth noting that these observations may not fit in our setting completely. Nonetheless, we have added discussion to page 10 “Against all methods”.

---

### Decision · Action_Editor_xnga · 2024-04-26

**Recommendation:** Accept as is

**Comment:**

The paper presents an alternative way to do regularization in offline RL and two new algorithms to use the technique. The reviewers all agreed that this is a significant contribution. The listed weaknesses were almost entirely about structure, presentation, readability, and technical clarity. The reviewers gave several points of detailed feedback on how to improve the paper, which the authors have taken to apply significant changes to the paper. As a result, the paper has greatly improved. All of the reviewers made note of the significant improvements and agreed upon an acceptance recommendation after the detailed changes.

**Audience:**

The audience is appropriate for TMLR

**Claims And Evidence:**

The claims are well-supported by the evidence.